# Deep Complex Spatio-Spectral Networks with Complex Visual Inputs

## Abstract

Complex-valued neural networks have attracted growing attention for their ability to handle complex-valued data with enhanced representational capacity. However, their potential in computer vision remains relatively untapped. In this paper, we introduce Deep Complex Spatio-Spectral Network (DCSNet), a fully complex-valued token-based, end-to-end neural network designed for binary segmentation tasks. Additionally, our DCSNet encoder can be used for image classification in the complex domain. We also propose an invertible real-to-complex (R2C) transform, which generates two complex-valued input channels, complex intensity and complex hue, while producing complex-valued images with distinct real and imaginary components. DCSNet operates in both spatial and spectral domains by leveraging complex-valued inputs and complex Fourier transform. As a result, the complex-valued representation is maintained throughout DCSNet, and we avoid the information loss typically associated with Real↔Complex transformations. Extensive experiments show that DCSNet surpasses existing complex-valued methods across various tasks on both real and complex-valued data and achieves competitive performance compared to existing real-valued methods, establishing a robust framework for handling both data types effectively.

## 1 Introduction

In the evolving landscape of deep learning, complex-valued neural networks (iCNNs) have shown great potential by enabling richer and more expressive representations. Despite their promise, iC-NNs remain relatively underinvestigated in addressing key computer vision problems. Addressing problems problems like binary segmentation necessitates a prominent understanding of the global and local context in the input. Despite complex-valued networks showing promising outcomes Löwe et al.; Stanic et al. (2023); Singhal et al. (2022), the application of complex-valued networks remains understudied due to a lack of complex-valued input and suboptimal complex-valued architectures. When tackling these issues, we are faced with a few challenges that need to be addressed: (i) A suitable way is required to have complex-valued inputs from real-valued RGB images. (ii) No prior work has shown promising results while maintaining the complex-valued nature of the input throughout different tasks. (iii) For binary segmentation, there is no objective function to handle complex-valued output.

Despite the challenges at hand, our motivation to explore complex-valued representation for computer vision tasks stems from the success demonstrated in diverse domains where complex-valued inputs are readily available, such as MRI Cole et al. (2021); Vasudeva et al. (2022), radar signals Gao et al. (2018); Georgiou & Koutsougeras (1992), and audio signals Hayakawa et al. (2018); Hu et al. (2020). Moreover, studies conducted in Arjovsky et al. (2016); Danihelka et al. (2016); Jojoa et al. (2022); Löwe et al., highlight the superior performance of complex-valued neural networks (iCNNs) compared to their real-valued counterparts. Additionally, iCNNs exhibit biological inspiration Reichert & Serre (2014) and greater generalization capacity Hirose & Yoshida (2012). Recent works Stanic et al. (2023); Halimeh & Kellermann (2022) also highlight the importance of complex-valued representation in images as well as audio, further fueling our exploration of iCNNs for computer vision tasks.

More specifically, we develop a complex-valued color transform R2C (real-to-complex), which converts real-valued images to complex-valued ones. We observe that any color vector in RGB space

Figure 1: We introduce a novel token-based Deep Complex Spatio-Spectral Networks (DCSNet), which leverage (i) complex R2C transform for producing complex-valued inputs, (ii) Fourier Filter module for capturing global context using complex Fourier transform, (iii) Complex T2T for progressively reducing tokens and its inverse Complex RT2T, and (iv) our novel complex-valued objective function $\mathcal{L}_{i\text{dense}}$ to optimize complex-valued output for binary segmentation.

can be projected to a grayscale line. The shortest angle between the two can provide us with the projection of color vector (on grayscale line) and the deflection ($\perp$ to grayscale line). Interestingly, the projection vector and the deflection vector are orthogonal, which creates an argand plane representing a complex number with the projection as the real part and deflection as the imaginary part. Similarly, we observe that the plane orthogonal to the grayscale line containing the color point can be considered an argand plane. Using the polar form, we can locate the point as a complex number using the perpendicular distance of the color point from grayscale line and a reference angle. Following this, we can generate two complex-valued representations for any color.

Furthermore, we introduce DCSNet, a novel end-to-end deep complex spatio-spectral network designed to operate on complex-valued images generated through the R2C transformation. It is first token-based approach utilizing complex-valued representation. The core component of DCSNet is the Fourier Filter Module, which transforms complex-valued tokens from the spatial domain to the frequency domain, applies a learnable Fourier filter, and subsequently maps the filtered results back to the spatial domain. Note that complex Fourier transform provides positive and negative frequency representation of complex-valued input. This architecture enables the network to capture both spatial and spectral domain information effectively, leading to preserved complex-valued representations. For binary segmentation tasks, we further propose a novel objective function $\mathcal{L}_{i\text{dense}}$ that optimizes the separation of foreground and background in the predicted complex-valued outputs. This is achieved by decomposing the output into real and imaginary components, enabling more effective supervision and improved performance in handling complex-valued predictions. We give an overview of our proposed approach in Fig. 1. Our extensive experiments reveal that our DCSNet outperforms all existing complex-valued methods for binary segmentation tasks on both real-valued and complex-valued datasets. Since we needed a backbone trained on a large dataset, we trained the encoder of our DCSNet on ImageNet-1k. Although our primary goal is not image classification, we observed that our encoder outperformed existing complex-valued methods for it as well, on both real-valued and complex-valued datasets.

Our contributions in this paper are as follows: (i) We propose R2C (real-to-complex), a novel complex-valued color transformation. (ii) We propose first token-based complex-valued network, DCSNet, which maintains the complex-valued information throughout. (iii) We propose a loss minimization strategy to handle complex-valued dense outputs. (iv) Our experimental analysis shows that our approach significantly improves for real and complex-valued data over existing methods across multiple tasks.

## 2 RELATED WORK

**Complex-valued Deep Learning:** The integration of complex numbers as weights in deep learning introduces novel possibilities for delving into two-dimensional spectra, as highlighted in previous works Tygert et al. (2016); Hirose & Yoshida (2011). Moreover, the significance of phase information in the firing rate of neurons is underscored by studies such as Reichert & Serre (2014) and Jiang et al. (2019), emphasizing the potential advantages of employing complex-valued representations in neural networks. Specifically, the observed behavior of synchronized neurons with similar

phases firing together, contrasted with asynchronous neurons with differing phases causing interference, bears a closer resemblance to the dynamics of biological neurons. This synchronization of inputs through neurons draws parallels to the gating mechanism found in both deep feedforward and recurrent neural networks Srivastava et al. (2015); Van den Oord et al. (2016); Kim & Adalı (2003).

Recent studies have showcased the superior generalization capacity of complex-valued networks, as evidenced by prior research Jojoa et al. (2022); Hirose & Yoshida (2012); Singhal et al. (2022). Notably, complex-valued autoencoders have outperformed slot-attention in the domain of object-centric learning Löwe et al. (2022); Stanic et al. (2023). Moreover, the application of complex-valued networks has proven beneficial in diverse areas, including saliency prediction Jiang et al. (2019; 2020) and iris recognition Nguyen et al. (2022). The findings presented in Cheung et al. (2019) further support the notion that employing a complex-valued vector can enhance the learning process for addressing multiple tasks.

**Fourier Transform in Vision:** The application of Fourier transform has been a cornerstone in digital image processing for decades, as acknowledged by seminal works in the field Gonzalez (2009); Pitas (2000). With the advent of Convolutional Neural Networks (CNNs) revolutionizing vision tasks He et al. (2016); Krizhevsky et al. (2012), there is a growing body of research integrating Fourier transform into deep learning methodologies Ding et al. (2017); Lee et al. (2018); Li et al. (2020); Yang & Soatto (2020). Some approaches employ discrete Fourier transform to transition images into the frequency domain, leveraging frequency information to enhance task performance Coates et al. (2011); Yang & Soatto (2020). Others exploit the convolution theorem, employing fast Fourier transform (FFT) to accelerate CNNs Ding et al. (2017); Li et al. (2020). In this study, we introduce a novel methodology using learnable Fourier filters to learn the global context in the Fourier domain, drawing inspiration from frequency filters in digital image processing Pitas (2000). Additionally, we capitalize on specific properties of FFT to reduce computational costs and parameter count.

## 3 PROPOSED METHOD

### 3.1 R2C: COMPLEX-VALUED COLOR TRANSFORM

In order to obtain complex-valued images from a real-valued one, our goal is to define a transformation $T : \mathbb{R}^{d_1} \to \mathbb{C}^{d_2}$, where $d_1$ & $d_2$ are dimensions of the input and output respectively. Since Real-valued images are typically in RGB format, we have $d_1 = H \times W \times 3$, where $H$, and $W$ are the height and width of the image. So, our target transformation function becomes $T : \mathbb{R}^{H \times W \times 3} \to \mathbb{C}^{H \times W \times k}$, where k is the number of complex channels in the complex image.

Defining the transformation function on the pixel level is relatively more straightforward. Let us consider $\hat{T}(p) : \mathbb{R}^3 \to \mathbb{C}^k, \forall p \in I_{RGB}$. If we find $k$ for each pixel $p$, we will ultimately have our transformation $T$. Each pixel $p$ in the image $I_{RGB}$ has its corresponding R, G & B values: $I_{RGB}(p) = \{I_r(p), I_g(p), I_b(p)\}$. Note that all the pixels in $I_{RGB}$ image are located in the three-dimensional RGB space. Given their corresponding $I_r, I_g \& I_b$ values, one can quickly locate them in this space. We use properties of RGB space and simple linear algebra to create a complex-valued representation of the image.

In RGB space (Fig. 2), we consider an isotropic vector $O$ (grayscale line) passing through the origin $C$ and making equal angles with each axis. For a given pixel $p$ in this space, we have a plane $P$, which has $O$ as a normal vector intersecting at point $E$ and contains pixel $p$ at point $F$. Also, The plane $P$ intersects the red, blue, and green axes at $B, A$, and $D$. Let us try to determine $k$ for pixel $p$.

### 3.1.1 COMPLEX INTENSITY CHANNEL ($I_\theta$):

It is crucial to notice that pixels lying on the vector $O$ will represent grayscale color since $I_r(p) = I_g(p) = I_b(p)$. If we project other pixels on vector $O$, we can obtain a projected grayscale version of the image $I_{rgb}$.

Taking this observation into account, in Fig 2, we see that vector $v = \overrightarrow{CF}$ for pixel $p$ makes an angle $\theta = \angle ECF$. $v$ has two orthogonal components in the direction of $\overrightarrow{CE}$(projection) & $\overrightarrow{EF}$(deflection). Using $\theta$ and $||v||$, we can decompose $v$ into its two orthogonal components, form-

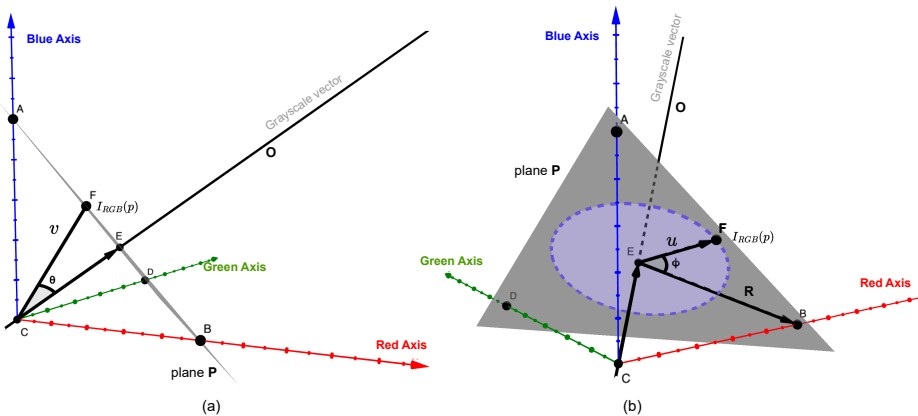

Figure 2: Proposed R2C color transformation: Given an RGB color $I_{RGB}(p) = \{I_R(p), I_G(p), I_B(p)\}$ for a pixel $p$. We have vectors $v = \overrightarrow{CF}$ & $u = \overrightarrow{EF}$. (a) Using $\theta$, we project $v$ onto the grayscale vector $O$, giving us projection $||v|| \cos \theta$ and deflection $||v|| \sin \theta$, which gives us the real and imaginary components of the first complex number. (b) Using $\phi$ between the reference vector $R$ and vector $u$. In the argand plane P (assuming $R$ as the real axis), we locate $p$, giving us the second complex number.

ing real and imaginary components of a complex value. Real component of this complex value $I_\theta$ being $||v|| \cos \theta$ and imaginary component being $||v|| \sin \theta$. It results in the first complex value of pixel $p$ as follows:

$$I_\theta(p) = ||v||e^{i\theta} = ||v|| \cos \theta + i||v|| \sin \theta \tag{1}$$

Here, $||.||$ represents the norm of the vector.

### 3.1.2   COMPLEX HUE CHANNEL ($I_\phi$):

Note that in Fig 2, $u = \overrightarrow{EF}$ for pixel $p$ lies in a plane that is $\perp$ to $O$, denoted as $P$. Assuming $P$ as an argand plane, we can locate $p$ using a complex number. For this, we need a reference axis. In Fig 2, we take this as a reference vector $R = \overrightarrow{EB}$, from $E$ to the intersection of plane $P$ and the red axis, given as $B$. Here, we assume $P$ as an argand plane with $R$ as the real axis and $R^\perp$ as the imaginary axis.

If we find the $\angle FEB = \phi$, we can easily locate $p$ in this argand plane using polar coordinates $(||u||, \phi)$. For computing $\phi$, we first find the shortest angle $\phi'$ between $u$ and $R$ using $\cos \phi' = \frac{u.R}{||u||||R||}$. Now we define $\phi$ as:

$$\phi = \begin{cases} \phi' & \text{if } I_b \geq I_g \\ 2\pi - \phi' & \text{else} \end{cases} \tag{2}$$

Now, using the polar coordinate, we can easily locate $p$ in argand plane $P$. This leads to the second complex value of $p$ as:

$$I_\phi(p) = ||u||e^{i\phi} = ||u|| \cos \phi + i||u|| \sin \phi \tag{3}$$

### 3.1.3   $i$RGB INPUT:

From above, we found that $k = 2$ and established $\hat{T}(p) = \mathbb{R}^3 \to \mathbb{C}^{k=2}, \forall p \in I_{rgb}$. This results in a complex transformation $T : \mathbb{R}^{H \times W \times 3} \to \mathbb{C}^{H \times W \times 2}$ such that $T(I_{RGB}) = I_{iRGB}$. Using $T$, we can convert our real-valued image $I_{RGB}$ to complex-valued image $I_{iRGB}$ with two complex-valued color channels. We use that to form the set required for the complex representation of $I_{iRGB}(p)$ as follows:

$$I_{iRGB}(p) = \{I_\theta(p), I_\phi(p)\} \tag{4}$$

We can separate complex-valued image $I_{iRGB}$ into its real and imaginary component as: $I_{iRGB} = I_{re} + iI_{im}$, where $I_{re} = \{||v|| \cos \theta, ||u|| \cos \phi\}$ and $I_{im} = \{||v|| \sin \theta, ||u|| \sin \phi\}$. Algorithm 1

---

**Algorithm 1** R2C Color Transformation

---

**Input:** A Real-valued RGB Image, $I_{RGB}$
**Output:** A Complex-valued transformed Image, $I_{iRGB}$

   **for** $p \in I_{RGB}$ **do**                                        ▷ for each pixel in the image
      ▷ Find angle between pixel vector $v$ and vector $O$.
      $\cos\theta = \frac{O \cdot v}{||O|| \times ||v||}$
      $\sin\theta = \sqrt{1 - \cos\theta}$
      $I_\theta(p) = ||v||(\cos\theta + i\sin\theta)$                             ▷ Intensity channel
      ▷ Find angle between between vector $u$ and vector $R$.
      $\phi' = \cos^{-1}(\frac{R \cdot u}{||R|| \times ||u||})$
      $I_{hsv} = rgb2hsv(I_{rgb})$
      **if** $\sin(hue(I_{hsv})) \geq 0$ **then**
          $\phi = \phi'$
      **else**
          $\phi = 2\pi - \phi'$
      **end if**
      $I_\phi(p) = ||u||(\cos\phi + i\sin\phi)$                           ▷ Color channel
   **end for**
   **return** $I_{iRGB} = \{I_\theta, I_\phi\}$                                 ▷ Final complex image

---

presents the algorithm for the above-presented R2C transform. Note that R2C is also invertible; we provide a detailed explanation for inverse R2C in Appendix A. Moreover, we considered using the Fourier transform to get complex input, but the complex input generated from the Fourier transform proved unsuitable. We present empirical results and discuss them in Appendix B.

## 3.2 DCSNet: Deep Complex Spatio-Spectral Networks

Recent advances in transformers Dosovitskiy et al. (2021); Yuan et al. (2021) demonstrate that self-attention based models can achieve good performances in solving various tasks. Following this approach, complex-valued transformer-based approaches Eilers & Jiang (2023); Yang et al. (2020); Dong et al. (2021) try to utilize self-attention. However, due to the nature of complex domain, they suffer from increased computation and poor results for image-related tasks. The proposed DCSNet architecture removes self-attention in favor of Fourier filters, enabling the model to retain information entirely within the complex domain while preserving global contextual information. As illustrated in Figure 3, DCSNet takes a complex input of size $H \times W$, divides it into patches, and unfolds these patches into tokens. The token length is progressively reduced in the spatial domain, while the Fourier filter operates in the frequency domain. This process is reversed in the decoder to progressively increase the token length.

## 3.3 DCSNet Encoder

We propose an encoder part that learns to generate image embeddings that can be used to generate binary segmentation output. However, unlike previous methods, we cannot use existing pre-trained encoders because we propose a new architecture that utilizes complex-valued information end-to-end. Hence, we also train DCSNet encoder on image classification, more details in experiments.

Initially, we have our complex input image $I \in \mathbb{C}^{H \times W \times 2}$. We apply complex convolution and generate complex-valued patches of size $= \frac{H}{4} \times \frac{W}{4}$. We introduce a Complex T2T module, which reduces the number of complex tokens, and a Fourier filter module, which acts as a global convolution operation in the frequency domain while maintaining the complex-valued nature of the input.

### 3.3.1 Fourier Filter module

We propose a Fourier filter module consisting of a Fourier filter and a complex MLP layer. Before each layer, we apply layer normalization. In the Fourier filter layer, given the input feature $x \in \mathbb{C}^{H \times W \times D}$, we first perform 2D DFT (see Appendix C) along the spatial dimension to convert $x$ to the frequency domain $X = F[x] \in \mathbb{C}^{H \times W \times D}$,

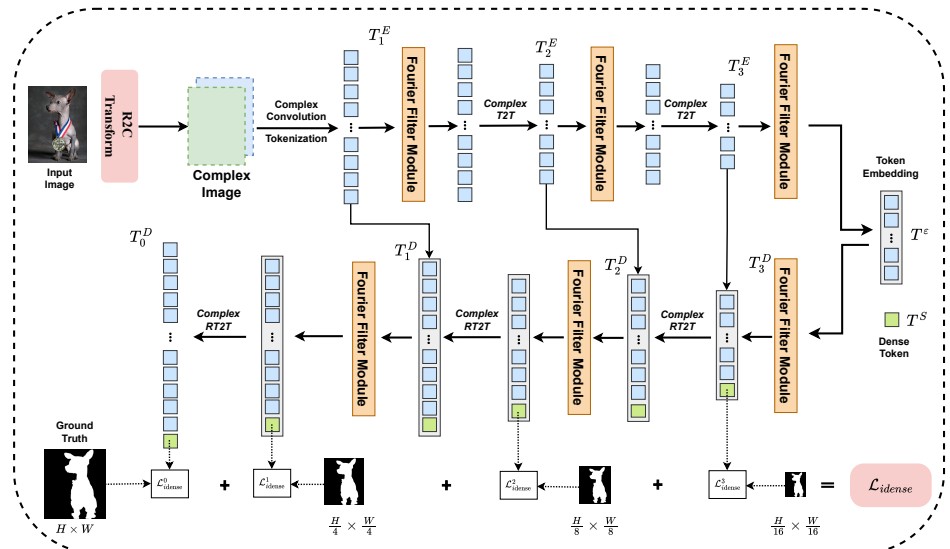

Figure 3: The overall architecture of our proposed DCSNet. It first encodes input image patch sequences to generate tokens to multiple resolutions ($T_1^E$, $T_2^E$, $T_3^E$), using Complex T2T. Then, we add a saliency token ($T^S$) to the output ($T^\varepsilon$) of the encoder. Finally, the decoder progressively upsamples the tokens using Complex RT2T while predicting saliency map at each step. We also optimize the generated map at each step using our proposed loss to improve the predicted map.

where $F[\cdot]$ denotes 2D complex DFT. The output $X$ of DFT is a complex tensor and represents $x$ in the frequency domain. We can now apply a learnable filter $K \in \mathbb{C}^{H \times W \times D}$ to $X$ by simple element-wise multiplication.

$$\hat{X} = K \odot X \qquad (5)$$

where $\odot$ denotes element-wise mul-

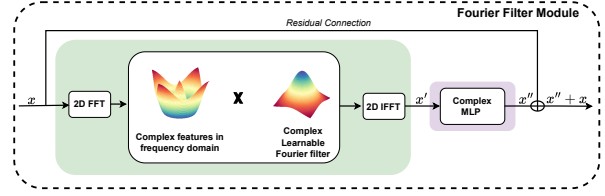

Figure 4: Fourier Filter Module takes a complex-valued input, applies a learnable complex filter in the frequency domain, and gives a complex-valued output.

tiplication. Since $K$ has the same dimension as $X$, it will act as a global filter in the frequency domain. Finally, we use inverse DFT to transform back to the spatial domain as $x' = F^{-1}[\hat{X}]$. We illustrate this process in Fig. 4. Fourier filter in the frequency domain is equivalent to a convolution with filter size $H \times W$ in the spatial domain. Hence, unlike convolution in the spatial domain, which focuses on local features due to the small filter size, the Fourier filter module focuses on the global context as shown by Rao et al. (2021). Moreover, keeping the operation in a complex domain helps us preserve additional complex information and both positive and negative frequencies in the spatial frequency domain.

### 3.3.2 COMPLEX T2T MODULE

Given a sequence of patch tokens $T$ with length $l$ from the previous layer, following T2T-ViT in Yuan et al. (2021), we introduce and iteratively apply the Complex T2T module (Fig. 5), which is composed of a re-structurization step and a soft split step, to model the local structure information in $T$ and obtain a new sequence of tokens.

This module consists of a structurization and a de-structurization step. In structurization step, $T \in \mathbb{C}^{l \times c}$ is reshaped to a 2D image $I \in \mathbb{C}^{h \times w \times c}$, where $l = h \times w$, to recover spatial structures. After the structurization step, we apply the de-structurization step, where $I$ is first split into $k \times k$ patches with $s$ overlapping. $p$ zero-padding is also utilized to pad image boundaries. Then, the image patches are unfolded to a sequence of tokens $T' \in \mathbb{C}^{l' \times ck^2}$, where the sequence length $l'$ is computed as $l' = h' \times w' = \lfloor \frac{h+2p-k}{k-s} + 1 \rfloor \times \lfloor \frac{w+2p-k}{k-s} + 1 \rfloor$.

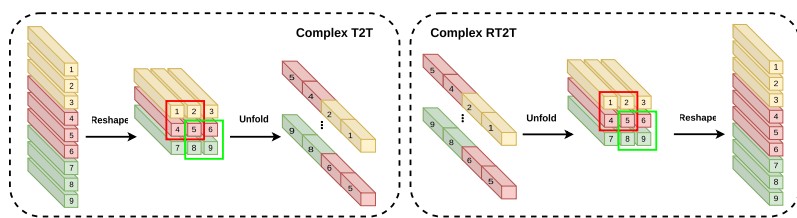

Figure 5: Complex T2T and Complex RT2T reduce and increase the number of tokens, respectively, while maintaining the complex nature of tokens.

### 3.4 DCSNET DECODER

As shown in Fig. 3, after obtaining complex feature embeddings $T^{\varepsilon}$, we want to predict the respective salient masks $M \in \{0,1\}^{H \times W \times 1}$ for each input $I$. The encoded length of $T^{\varepsilon}$ is quite small $l^{\varepsilon} = [\frac{H}{16}, \frac{W}{16}]$. We use the reverse version of the Complex T2T module, Complex RT2T. Complex RT2T upsamples each token into multiple subtokens while maintaining common information between them. We leverage low-level tokens from the encoder to provide additional information simultaneously. Progressively, we upsample embedded tokens $T^{\varepsilon}$ and add information from $T_1$ and $T_2$ using concatenation and complex linear projection.

#### 3.4.1 COMPLEX RT2T MODULE

In order to upsample the encoded information in tokens, we introduce the reverse version of the complex T2T module. Specifically, we first project the input patch tokens to reduce their embedding dimension from $d = 384$ to $c = 64$. Then, we use another complex linear projection to expand the embedding dimension from $c$ to $ck^2$. Next, similar to the de-structurization step in complex T2T, each token is seen as a $k \times k$ image patch, and neighboring patches have $s$ overlapping. Then, we can fold the tokens as an image using $p$ zero-padding. Finally, we reshape the image to match the upsampled tokens, as shown in Fig. 5.

#### 3.4.2 TOKEN-BASED MASK GENERATION

Inspired by existing transformer architecture Dosovitskiy et al. (2021); Yuan et al. (2021); Eilers & Jiang (2023); Liu et al. (2021), we add a dense token as shown in Fig. 3 In doing so, we design a complex-valued dense token $T^S \in \mathbb{C}^{1 \times d}$, where $d$ is the embedding dimension. At each level in the decoder, we add $T^S$ to the encoded patch tokens $T_i^D, i \in \{0, 1, 2, 3\}$. When the modified tokens are processed through the Fourier filter module, the dense token learns dense information from an image by progressively extracting feature information from other tokens. For binary segmentation, we do not use any self-attention module. We again send our obtained tokens $T_1^D$ to a Fourier filter module, and the third complex RT2T module upsamples the tokens from $1/4$ to full resolution.

### 3.5 OBJECTIVE FUNCTION

Since our architecture is complex-valued, we must ensure that the loss function can handle the complex output. For classification tasks, we employ the complex loss function proposed by Yadav & Jerripothula (2023). However, existing loss functions cannot handle complex outputs directly for binary segmentation. To tackle this problem, we propose a modified binary cross entropy loss $\mathcal{L}_{i\text{dense}}$. We obtain a complex-valued output $z_j$ from our network for $j^{th}$ input. Given the real-valued ground truth $y_j$, we construct a complex-valued ground truth by considering the foreground dense map as the real part and the background dense map as the imaginary part, as shown in Fig. 6. It turns out to be a complex-valued map with the foreground pixel having value = 1 and the background pixel having value = $i$, just like real-valued ground-truth dense maps have foreground pixel value = 1 and background pixel value = 0. We develop a loss $\mathcal{L}_{i\text{dense}}$ which can minimize the comlpex-valued output of our network $z_j$ with help of complex-valued ground truth saliency map $y_j + i(1 - y_j)$.

We formulate our $\mathcal{L}_{i\text{dense}}$ as follows:

$$\mathcal{L}_{i\text{dense}}(z_j) = \sum_{j=1}^{N} \sum_{k=0}^{3} \mathcal{L}_{BCE}(y_j^k, \Re(z_j^k)) + \mathcal{L}_{BCE}((1 - y_j^k), \Im(z_j^k)) \tag{6}$$

where $\mathcal{L}_{BCE}$ is binary cross entropy loss, $k$ indicates token levels $T_k^D$ in decoder, $N$ is total number of input images, and $\Re$ and $\Im$ are real and imaginary components of complex output $z_j$. To generate the final binary mask from complex-valued output $z_j$, we take the average of real and complement of the imaginary component as $mask = (\Re(z_j) + (1 - \Im(z_j)))/2$.

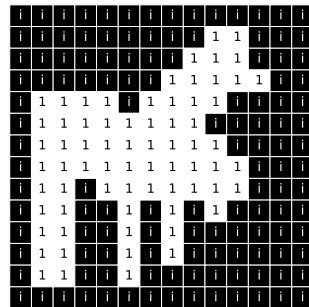

Figure 6: Complex grountruth, foreground pixels are denoted by 1, and background pixels as 0

## 4 EXPERIMENTS

In this section, we present the evaluation of our proposed Deep Complex Spatio-Spectral Networks (DCSNet) and compare their performance with other methods (both real and complex-valued) for multiple task.

### 4.1 DATASETS & METHODS

We conduct extensive experiments on binary segmentation tasks. We present results on both real-valued and complex-valued datasets.

**Real-valued Datasets:** For binary segmentation, we assess the performance of DCSNet on salient object detection, defocus blur detection, and shadow detection using multiple datasets. Furthermore, we provide a comprehensive comparison of real-valued and complex-valued methods across these tasks. In the process, we had to train our backbone/encoder on a large dataset such as ImageNet-1k Deng et al. (2009). We also used CIFAR10 dataset Krizhevsky et al. for running our encoder-related ablation studies.

We follow the approach outlined in Liu et al. (2021) for comparison with real-valued methods on salient object detection. We evaluate our model on five widely-used benchmark datasets: ECSSD Yan et al. (2013) (1,000 images), PASCAL-S Li et al. (2014) (850 images), HKU-IS Li & Yu (2015) (4,447 images), DUT-O Yang et al. (2013) (5,168 images), and DUTS Wang et al. (2017) (10,553 images). We use four commonly employed evaluation metrics across these datasets: $S_m$ Fan et al. (2017), maxF (maximum F-measure), $E_\xi^{max}$ Fan et al. (2018) (maximum enhanced-alignment measure), and MAE (mean absolute error).

We use the CUHKShi et al. (2014) and DUT Zhao et al. (2018) datasets for defocus blur detection. The CUHK dataset contains 704 defocus images, while DUT comprises 1,100 images. Our method is trained on 604 images from CUHK and 600 from DUT. We evaluate the remaining test images from both datasets using F-measure ($F_\beta$) and mean absolute error (MAE) as evaluation metrics. For shadow detection, we assess our model on the SBU Vicente et al. (2016) and ISTD Wang et al. (2018) datasets. We train the model separately on each dataset and report the performance using the balanced error rate (BER).

**Complex-valued Datasets:** We utilize the Simulated InSAR Building dataset Chen (2020) for building detection, treating the layover class as the foreground. The dataset consists of 270 training images and 42 test images. We use mean Intersection over Union (mIoU) as the evaluation metric and compare our results with other complex-valued methods. We also benchmarked our encoder on two complex-valued datasets: MSTAR Ross et al. (1998) and S1SLC_CVDL Mohammadi Asiyabi et al. (2023) for classification tasks.

We resize input images to $256 \times 256$ and then randomly crop $224 \times 224$ image regions as the model input and use random flipping as data augmentation. We randomly initialize weights, set the batch size to 8, and use Adam Kingma & Ba (2015) optimizer. For implementating complex-valued operations, we use the PyTorch library which provides native support for complex tensor and complex operations including convolution and complex FFT.

Table 1: Comparing classification accuracy(%) of our encoder with only other complex-valued method to show results on Real-valued (ImageNet-1kDeng et al. (2009)) and Complex-valued (MSTARRoss et al. (1998), S1SLC_CVDLMohammadi Asiyabi et al. (2023)) datasets.

(a) Real-valued Dataset

| Model | Top-1 Acc. (%) |
| --- | --- |
| ResNet50He et al. (2016) | 76.1 |
| Vit-S/16Dosovitskiy et al. (2021) | 78.1 |
| GFNet-xsRao et al. (2023) | 78.6 |
| ResMLP-12Touvron et al. (2022) | 76.6 |
| ResNet152 (DCN)Trabelsi et al. (2018) | 72.6 |
| ResNet152 (FCCN)Yadav & Jerripothula (2023) | 77.3 |
| DCSNet Encoder | **78.8** |

(b) Complex-valued Dataset

| Dataset | DCN | FCCN | DCSNet Encoder |
| --- | --- | --- | --- |
| MSTAR | 96.1 | 97.4 | **97.7** |
| S1SLC_CVDL | 93.2 | 89.8 | **91.6** |

Table 2: Quantitative comparison of our proposed DCSNet with other real-valued RGB SOD methods on five benchmark datasets. "-R" and "-R2" means the ResNet50 and Res2Net backbone, respectively. The values in red are best, and the ones in blue are second best. Our DCSNet performs best or second best 55% of the time.

| Method | Param(M) | DUTSWang et al. (2017) | | | | ECSSDYan et al. (2013) | | | | HKU-ISLi & Yu (2015) | | | | PASCAL-SLi et al. (2014) | | | | DUT-OYang et al. (2013) | | | | Average | | | |
| --- | --- | --- | --- | --- | --- | --- | --- | --- | --- | --- | --- | --- | --- | --- | --- | --- | --- | --- | --- | --- | --- | --- | --- | --- | --- |
| | | $S_m$ ↑ | maxF↑ | $E_\xi^{max}$ ↑ | MAE↓ | $S_m$ ↑ | maxF↑ | $E_\xi^{max}$ ↑ | MAE↓ | $S_m$ ↑ | maxF↑ | $E_\xi^{max}$ ↑ | MAE↓ | $S_m$ ↑ | maxF↑ | $E_\xi^{max}$ ↑ | MAE↓ | $S_m$ ↑ | maxF↑ | $E_\xi^{max}$ ↑ | MAE↓ | $S_m$ ↑ | maxF↑ | $E_\xi^{max}$ ↑ | MAE↓ |
| Real-valued | | | | | | | | | | | | | | | | | | | | | | | | | |
| PiCANetLiu et al. (2018) | 47.22 | 0.863 | 0.840 | 0.915 | 0.040 | 0.916 | 0.929 | 0.953 | 0.035 | 0.905 | 0.913 | 0.951 | 0.031 | 0.846 | 0.824 | 0.882 | 0.072 | 0.826 | 0.767 | 0.865 | 0.054 | 0.871 | 0.854 | 0.913 | 0.046 |
| BASNetQin et al. (2019) | 87.06 | 0.866 | 0.838 | 0.902 | 0.047 | 0.916 | 0.931 | 0.951 | 0.037 | 0.909 | 0.919 | 0.952 | 0.032 | 0.837 | 0.819 | 0.868 | 0.083 | 0.836 | 0.779 | 0.872 | 0.057 | 0.872 | 0.856 | 0.909 | 0.051 |
| PoolNetLiu et al. (2019) | 68.26 | 0.879 | 0.853 | 0.917 | 0.041 | 0.917 | 0.929 | 0.948 | 0.042 | 0.916 | 0.920 | 0.955 | 0.032 | 0.852 | 0.830 | 0.880 | 0.076 | 0.832 | 0.769 | 0.869 | 0.056 | 0.879 | 0.864 | 0.914 | 0.049 |
| EGNet-RZhao et al. (2019a) | 111.64 | 0.887 | 0.866 | 0.926 | 0.039 | 0.925 | 0.936 | 0.955 | 0.037 | 0.0918 | 0.923 | 0.956 | 0.031 | 0.852 | 0.825 | 0.874 | 0.080 | **0.841** | 0.778 | 0.878 | **0.053** | 0.886 | 0.868 | 0.918 | 0.048 |
| MINet-RPang et al. (2020) | 162.38 | 0.884 | 0.864 | 0.926 | **0.037** | 0.925 | 0.938 | 0.957 | **0.034** | 0.919 | 0.926 | 0.960 | 0.031 | 0.856 | 0.831 | 0.883 | 0.071 | 0.833 | 0.769 | 0.869 | 0.056 | 0.885 | 0.868 | 0.919 | 0.046 |
| LDF-RWei et al. (2020) | 25.15 | 0.892 | **0.877** | 0.930 | **0.034** | 0.925 | 0.938 | 0.954 | **0.034** | 0.920 | **0.929** | 0.958 | **0.028** | 0.861 | **0.839** | 0.888 | **0.067** | 0.839 | **0.782** | 0.879 | **0.052** | 0.892 | 0.876 | 0.921 | **0.043** |
| CSF-R2Gao et al. (2020) | 36.53 | 0.890 | 0.869 | 0.929 | **0.037** | **0.931** | 0.942 | **0.960** | **0.033** | - | - | - | - | 0.863 | **0.839** | 0.885 | 0.073 | 0.838 | 0.775 | 0.869 | 0.055 | 0.892 | 0.874 | 0.911 | 0.049 |
| GateNet-RZhao et al. (2020) | 128.63 | 0.891 | **0.874** | 0.932 | 0.038 | 0.924 | 0.935 | 0.955 | 0.038 | **0.921** | 0.926 | 0.959 | 0.031 | 0.863 | 0.836 | 0.886 | 0.071 | 0.840 | **0.782** | 0.878 | 0.055 | 0.888 | 0.874 | 0.922 | 0.047 |
| VSTLiu et al. (2021) | 44.48 | **0.896** | **0.877** | **0.939** | 0.037 | **0.932** | **0.944** | **0.964** | **0.034** | **0.928** | 0.937 | **0.968** | 0.030 | **0.873** | **0.850** | **0.900** | **0.067** | **0.850** | **0.800** | **0.888** | 0.058 | **0.904** | **0.894** | **0.932** | 0.045 |
| Complex-valued | | | | | | | | | | | | | | | | | | | | | | | | | |
| FCCNYadav & Jerripothula (2023) | 48.61 | 0.811 | 0.750 | 0.872 | 0.070 | 0.874 | 0.876 | 0.918 | 0.082 | 0.867 | 0.857 | 0.919 | 0.058 | 0.807 | 0.783 | 0.854 | 0.100 | 0.787 | 0.695 | 0.827 | 0.079 | 0.824 | 0.813 | 0.878 | 0.078 |
| SCVUNetWei et al. (2023) | 54.15 | 0.824 | 0.769 | 0.874 | 0.063 | 0.882 | 0.887 | 0.922 | 0.062 | 0.877 | 0.872 | 0.929 | 0.052 | 0.815 | 0.795 | 0.865 | 0.089 | 0.788 | 0.697 | 0.875 | 0.077 | 0.831 | 0.827 | 0.893 | 0.069 |
| DCSNet (ours) | 39.12 | **0.894** | **0.874** | **0.941** | 0.039 | **0.927** | **0.945** | **0.960** | **0.034** | 0.917 | 0.924 | **0.967** | **0.029** | **0.866** | **0.850** | **0.903** | **0.062** | 0.839 | 0.776 | **0.880** | 0.056 | **0.893** | **0.895** | **0.930** | **0.044** |

## 4.2 RESULTS

**DCSNet encoder results:** Our proposed DCSNet encoder is the first token-based approach for complex-valued image classification. When comparing our encoder on image classification (Table 1a and 1b), we observe that DCSNet beats FCCNYadav & Jerripothula (2023) on both large-scale real-valued dataset (ImageNet) and complex-valued datasets (MSTARRoss et al. (1998)& S1SLC_CVDLMohammadi Asiyabi et al. (2023)). We provide additional comparisons with FCCN in Appendix E. We obtain significantly better results than Yadav & Jerripothula (2023) while maintaining fewer parameters, marking the best result for complex-valued image classification on large-scale datasets for both real and complex-valued datasets.

**Comparison on binary segmentation:** To show the applicability and efficiency of our proposed method, we also compare our DCSNet with 9 other real-valued salient object detection methods on five different datasets for a more extensive comparison with real-valued methods. We also present results and compare them with two recent complex-valued methods: FCCN & SCVUNet. For FCCN, we follow a CNN-based encoder-decoder approach for binary segmentation, while SCVUNet is directly utilized for the real-valued dataset. We present our results in Table 2, which shows that DCSNet performs either best or second best **55**% of the time. We also present qualitative results in Appendix F. Similarly, for both defocus blur detection, and shadow detection, we outperform existing real-valued and complex-valued methods. We present the comparison on both tasks in Tab. 3& 4.

When comparing complex-valued data for foreground extraction (Tab. 5), we see a similar pattern, i.e., DCSNet outperforms existing complex-valued methods decisively. Our proposed fully complex-valued method obtains results comparable to existing real-valued methods. It marks the first milestone for the application of complex-valued methods in binary segmentation tasks.

## 4.3 ABLATION STUDY

We conduct three ablation studies to highlight the importance of our contributions. The first two ablation studies are conducted on four benchmark datasets for SOD. The third study is conducted on CIFAR10 following Yadav & Jerripothula (2023) for image classification. In addition to our encoder, we take a smaller version with fewer parameters to observe performance variation. All the models for the ablation study are trained from scratch in order to maintain fairness.

Table 3: Comparison with real and complex-valued methods on defocus blur detection. Red: best, Blue: second best.

| Method | Param (M) | DUT $\mathcal{F}_\beta\uparrow$ | DUT $\mathcal{M}\downarrow$ | CUHK $\mathcal{F}_\beta\uparrow$ | CUHK $\mathcal{M}\downarrow$ |
|---|---|---|---|---|---|
| Real-valued | | | | | |
| DeFusionNet Tang et al. (2019) | - | 0.823 | 0.118 | 0.818 | 0.117 |
| BTBNet Zhao et al. (2018) | - | 0.827 | 0.138 | 0.889 | 0.082 |
| CENet Zhao et al. (2019b) | - | 0.817 | 0.135 | 0.906 | 0.059 |
| DAD Zhao et al. (2021b) | 44.13 | 0.794 | 0.153 | 0.884 | 0.079 |
| EFENet Zhao et al. (2021a) | 43.61 | 0.854 | 0.094 | 0.914 | 0.053 |
| Complex-valued | | | | | |
| FCCN Yadav & Jerripothula (2023) | 48.61 | 0.860 | 0.104 | 0.898 | 0.081 |
| SCVUNet Wei et al. (2023) | 54.15 | 0.848 | 0.096 | 0.901 | 0.078 |
| DCSNet (ours) | 39.12 | 0.894 | 0.058 | 0.907 | 0.045 |

Table 4: Comparison with real and complex-valued approaches on shadow detection. Red: best, Blue: second best.

| Method | Param (M) | ISTD BER↓ | SBU BER↓ |
|---|---|---|---|
| Real-valued | | | |
| Stacked CNN Vicente et al. (2016) | - | 8.60 | - |
| BDRAR Zhu et al. (2018) | 42.45 | 2.69 | 3.89 |
| DSC Hu et al. (2018) | 122.49 | 3.42 | 5.59 |
| DSD Zheng et al. (2019) | 58.15 | 2.17 | 3.45 |
| MTMT Chen et al. (2020) | 44.12 | 1.72 | 3.15 |
| FDRNet Zhu et al. (2021) | - | 1.55 | 3.04 |
| Complex-valued | | | |
| FCCN Yadav & Jerripothula (2023) | 48.61 | 1.78 | 3.22 |
| SCVUNet Wei et al. (2023) | 54.15 | 1.91 | 3.25 |
| DCSNet (ours) | 39.12 | 1.49 | 3.05 |

Table 5: Comparison on Complex-valued Building InSAR dataset Chen (2020) for foreground extraction. Red: best, Blue: second best.

| Method | Param (M) | Building mIoU ↑ |
|---|---|---|
| FCCN Yadav & Jerripothula (2023) | 48.61 | 0.82 |
| SCVUNet Wei et al. (2023) | 54.15 | 0.86 |
| DCSNet (ours) | 39.12 | 0.89 |

**Effects of complex input on binary segmentation:** In this ablation study, we analyze the contribution of various inputs and channels in $I_{iRGB}$ input by providing them one at a time. Results are shown in Table 6. Each $I_{iRGB}$ channel performs well; however, we get the best results when both are taken together. Even while using other inputs to DCSNet, i.e., RGB and iHSV Yadav & Jerripothula (2023), we observe that our complex input performs better for binary segmentation.

**Effects of $\mathcal{L}_{i\mathbf{dense}}$ & $T^S$:** To analyze the importance of dense loss $\mathcal{L}_{i\text{dense}}$, we use binary cross entropy loss and only optimize the real component of complex-valued output. Similarly, to observe the importance of dense token $T^S$, we remove it from our model. We present the result of these ablations in Table 6.

Table 6: Results of ablation study that highlights the effect of both channels in $I_{iRGB}$ and various other inputs. We also highlight the importance of our loss $\mathcal{L}_{idense}$, and dense token($T^S$).

| Dataset | DUTS $S_m\uparrow$ | maxF↑ | $E_\xi^{max}\uparrow$ | MAE↓ | ECSSD $S_m\uparrow$ | maxF↑ | $E_\xi^{max}\uparrow$ | MAE↓ | PASCAL-S $S_m\uparrow$ | maxF↑ | $E_\xi^{max}\uparrow$ | MAE↓ | DUT-O $S_m\uparrow$ | maxF↑ | $E_\xi^{max}\uparrow$ | MAE↓ |
|---|---|---|---|---|---|---|---|---|---|---|---|---|---|---|---|---|
| baseline (VST) | 0.732 | 0.644 | 0.785 | 0.130 | 0.819 | 0.808 | 0.866 | 0.112 | 0.742 | 0.691 | 0.866 | 0.157 | 0.731 | 0.631 | 0.782 | 0.131 |
| ours (RGB) | 0.731 | 0.641 | 0.774 | 0.147 | 0.820 | 0.801 | 0.857 | 0.114 | 0.732 | 0.658 | 0.847 | 0.139 | 0.731 | 0.625 | 0.781 | 0.136 |
| ours (iHSV) | 0.735 | 0.648 | 0.780 | 0.128 | 0.824 | 0.813 | 0.864 | 0.105 | 0.742 | 0.710 | 0.813 | 0.128 | 0.738 | 0.631 | 0.789 | 0.112 |
| ours ($I_\theta$) | 0.729 | 0.629 | 0.790 | 0.123 | 0.810 | 0.791 | 0.865 | 0.109 | 0.743 | 0.701 | 0.794 | 0.144 | 0.735 | 0.634 | 0.792 | 0.122 |
| ours ($I_\phi$) | 0.726 | 0.631 | 0.791 | 0.120 | 0.805 | 0.788 | 0.866 | 0.112 | 0.745 | 0.692 | 0.789 | 0.139 | 0.729 | 0.636 | 0.785 | 0.119 |
| ours (w/o $\mathcal{L}_{idense}$) | 0.731 | 0.630 | 0.789 | 0.122 | 0.824 | 0.807 | 0.872 | 0.097 | 0.747 | 0.702 | 0.794 | 0.143 | 0.733 | 0.633 | 0.788 | 0.123 |
| ours (w/o $T^S$) | 0.734 | 0.649 | 0.787 | 0.113 | 0.825 | 0.818 | 0.878 | 0.091 | 0.749 | 0.709 | 0.703 | 0.138 | 0.730 | 0.646 | 0.792 | 0.114 |
| ours (iRGB) | 0.740 | 0.654 | 0.801 | 0.107 | 0.831 | 0.825 | 0.884 | 0.083 | 0.747 | 0.713 | 0.801 | 0.131 | 0.739 | 0.645 | 0.795 | 0.109 |

Table 7: Improvements over baseline two variants of GFNet Rao et al. (2021) -xs and -ti. Here, we study the results of another complex-valued input, iHSV Yadav & Jerripothula (2023), and the effect of each complex-valued channel in Image classification on the CIFAR-10 dataset.

| Model | GFNet-xs (RGB) | (HSV) | (iHSV) | ($I_\theta$) | ($I_\phi$) | (iRGB) | GFNet-ti (RGB) | (HSV) | (iHSV) | ($I_\theta$) | ($I_\phi$) | (iRGB) |
|---|---|---|---|---|---|---|---|---|---|---|---|---|
| | GFNet-xs | | DCSNet Encoder | | | | GFNet-ti | | DCSNet Encoder-small | | | |
| Acc (%) | 93.6 | 93.3 | 93.5 | 92.7 | 91.4 | **94.3** | 92.1 | 91.9 | 92.2 | 91.7 | 90.3 | **92.8** |

**Effects of complex input on image classification:** This study observes the effect of various complex-valued inputs and each channel of $I_{iRGB}$ for image classification. We compare these results with two variants of real-valued model GFNet Rao et al. (2021). Our DCSNet encoders contain a similar number of parameters as variants of GFNet. From Table 7, we can observe the role of proposed complex-valued input.

## 5 CONCLUSION

In this work, we have presented DCSNet, a fully complex-valued token-based network for binary segmentation tasks, which operates both in spatial and frequency domain. It takes complex-valued input generated from our R2C transform and optimizes the complex-valued dense output using our proposed loss function. While maintaining complex-valued information throughout, our model outperforms previous complex-valued methods on various tasks and both real and complex-valued data, presenting a robust approach using complex-valued representation.

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

## A   INVERTIBILITY OF R2C TRANSFORM

As mentioned in the main manuscript, R2C transform is invertible. Given the function transform function $T : \mathbb{R}^{H \times W \times 3} \to \mathbb{C}^{H \times W \times 2}$, we need to show that it is invertible, which implies it is a one to one function. In Algorithm. 2, we present a pseudo code for inverse R2C color transform. We focus on obtaining the vector $v = \overrightarrow{CF}$, as shown in Fig 7. Once it is obtained, we will have the location of pixel $p$ at F.

We provide a pseudo code for the inverse R2C transform. Given the complex image $I_{iRGB}$, we convert each pixel value from complex to real. First, we find $u = \overrightarrow{EF}$ (Fig. 7) using its components along $R$ and $R^\perp$, then using the component of $v$ along $O = \overrightarrow{CE}$, we obtain $v = \overrightarrow{CE} + \overrightarrow{EF}$ and hence the point F which is the location of pixel $p$ in RGB space. We also provide a code demo for R2C transform in the supplementary zip folder.

---

**Algorithm 2** Pseudo code for inverse R2C Color Transformation

---

**Input:** A complex-valued Image, $I_{iRGB}$
**Output:** A real-valued Image, $I_{RGB}$
  **for** $p \in I_{iRGB}$ **do**                                        ▷ for each pixel in the complex image
      ▷ Our goal is to find vector $v = \overrightarrow{CF} = \overrightarrow{CE} + \overrightarrow{EF}$
      ▷ Let us assume $u = \overrightarrow{EF}$, $O$ = grayscale vector
      ▷ We have two complex numbers for pixel $p$
      $C_1 = ||v|| \cos\theta + i||v|| \sin\theta$
      $C_2 = ||u|| \cos\phi + i||u|| \sin\phi$
      point E = $(\Re(C_1), \Re(C_1), \Re(C_1))$                             ▷ lies on $O$
      plane P = plane formed perpendicular to vector $O$ at point E
      point B = Intersection of Plane P and Red axis            ▷ plane and axis are known
      vector R = $\overrightarrow{EB}$
      vector $R^\perp$ = lies in plane P, and has angle $+\pi/2$ with R
      ▷ find vector u in RGB space
      $u_1 = \Re(C_2)\hat{R}$                                    ▷ component of $u$ along R
      $u_2 = \Im(C_2)\hat{R^\perp}$                           ▷ component of $u$ along $R^\perp$
      $u = \overrightarrow{EF} = u_1 + u_2$
      ▷ find component of vector $v$ in RGB space
      $\overrightarrow{CE} = \Re(C_1)\hat{O}$                              ▷ component of $v$ along O
      ▷ find vector $v$ in RGB space
      $v = \overrightarrow{CF} = \overrightarrow{CE} + \overrightarrow{EF} = [I_R(p), I_G(p), I_B(p)]$
      point $F = (I_R(p), I_G(p), I_B(p))$
  **end for**
  **return** $I_{RGB} = \{I_R, I_G, I_B\}$                                  ▷ Final real-valued image

---

## B   COMPLEX INPUT USING FOURIER TRANSFORM

In addition to the R2C transform, DFT (Discrete Fourier transform) can be used to generate complex-valued representations of images. However, due to a lack of spatial structure and localized context, Fourier representation does not provide additional benefits. To assert this observation, we conduct an experiment with two variations of the DCSNet encoder on the CIFAR10 dataset in Tab. 8.

Table 8: Comparison with DFT (Discrete Fourier transform) and iRGB input to DCSNet encoders.

| Input | DCSNet-Encoder | | DCSNet-Encoder small | |
|---|---|---|---|---|
| | DFT | iRGB | DFT | iRGB |
| Acc(%) | 85.8 | **94.3** | 77.6 | **92.8** |

Table 9: Additional ablation study to compare our Fourier filter module and self-attention.

| Metric | DUTS | | ECSSD | | HKU-IS | | PASCAL-S | | DUT-O | |
|---|---|---|---|---|---|---|---|---|---|---|
| | SA | Ours | SA | Ours | SA | Ours | SA | Ours | SA | Ours |
| $S_m \uparrow$ | 0.735 | **0.740** | 0.828 | **0.831** | **0.748** | 0.733 | 0.745 | **0.747** | 0.738 | **0.739** |
| maxF $\uparrow$ | 0.651 | **0.654** | 0.820 | **0.825** | 0.688 | **0.689** | 0.711 | **0.713** | **0.645** | **0.645** |
| $E_\xi^{max} \uparrow$ | 0.788 | **0.801** | 0.875 | **0.884** | 0.763 | **0.791** | 0.787 | **0.801** | 0.788 | **0.795** |
| MAE $\downarrow$ | **0.104** | 0.107 | 0.091 | **0.083** | 0.170 | **0.157** | 0.146 | **0.131** | 0.112 | **0.109** |

## C  BACKGROUND

We start by introducing the discrete Fourier transform (DFT), which plays a vital role in signal processing. For clarity we consider 1D DFT. Give a sequence of $N$ complex numbers $x[n], 0 \leq n \leq N - 1$, the DFT of $x[n]$ will be:

$$X[k] = \sum_{n=0}^{N-1} x[n]e^{-i(2\pi/N)kn} = \sum_{n=0}^{N-1} x[n]W_N^{kn} \tag{7}$$

where $i$ is the iota, and $W_N = e^{-i(2\pi/N)}$.

Since $X[k]$ repeats on intervals of length N, we can take value of $X[k]$ at $N$ consecutive points $k = 0, 1, \ldots, N - 1$. Specifically, $X[k]$ represents the spectrum of sequence $x[n]$ at the frequency $w_k = 2\pi k/N$.

It is well known that the DFT is a bijective function, i.e., the inverse of DFT function exists. Given $X[k]$, we can recover the original signal $x[n]$ by the inverse DFT also denoted as IDFT

$$x[n] = \frac{1}{N} \sum_{k=0}^{N-1} X[k]e^{i(2\pi/N)kn} \tag{8}$$

Note that, in real DFT, the input $x[n]$ is real, and its DFT is conjugate symmetric Rao et al. (2021), i.e., $X[N - k] = X^*[k]$. The reverse is true as well; if we perform IDFT to $X[k]$ which is conjugate symmetric, a real discrete signal can be covered. This is a major point that half of the DFT $\{X[k] : 0 \leq k \leq [N/2]\}$ contains full information of $x[n]$.

However, in complex DFT, the input $x[n]$ is complex. Hence, its DFT includes both positive and negative frequencies. This means that unlike real DFT, $X[k]$ is not conjugate symmetric. $X[k]$ between 0 to $N/2$ is positive and between $N/2$ and $N - 1$ its negative.

The DFT described above can be extended to 2D signals. Given 2D complex signal $X[m, n], 0 \leq m \leq M - 1, 0 \leq n \leq N - 1$, the 2D DFT of $x[m.n]$ is given by:

$$X[u, v] = \sum_{M-1}^{m=0} \sum_{N-1}^{n=0} x[m, n]e^{-i2\pi(\frac{um}{M} + \frac{vn}{N})} \tag{9}$$

The 2D DFT can be thought of as performing 1D DFT on the two dimensions alternatively. Similar to complex 1D DFT, 2D DFT does not have properties of conjugate symmetry.

## D  ABLATION EXPERIMENTS

We also provide another ablation study to compare complex-valued self-attention and our proposed learnable Fourier filter. For this ablation, we replace the Fourier filter module with the complex-valued self-attention (SA) module as proposed by Eilers & Jiang (2023); Yang et al. (2020). We can notice performance-decline in most cases when the Fourier filter module is replaced. Table 9 empirically validates our proposed method for capturing global information in complex-domain.

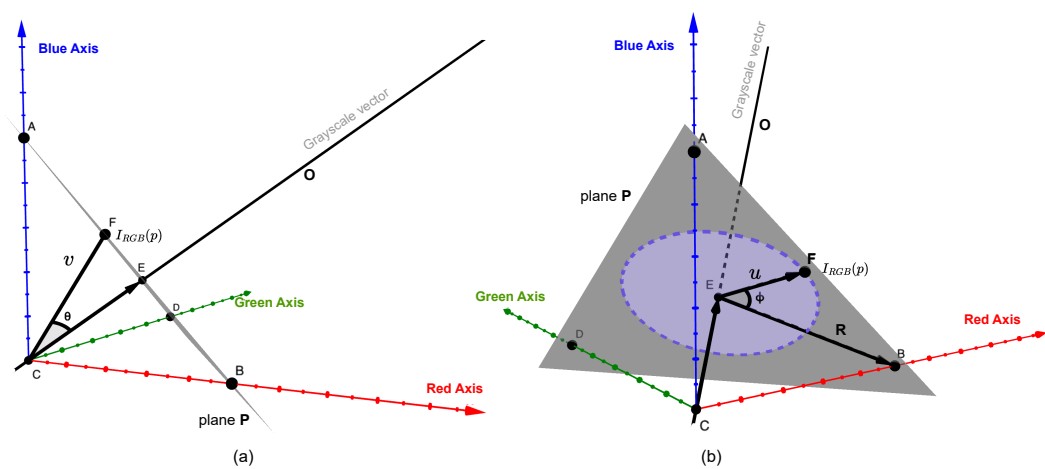

(a)                     (b)

Figure 7: R2C color transformation: Given an RGB color $I_{RGB}(p) = \{I_R(p), I_G(p), I_B(p)\}$ for a pixel $p$, we transform $p$ as $I_{iRGB} = \{I_\theta(p), I_\phi(p)\}$.

# E    ADDITIONAL RESULTS

For comparison with complex-valued method for saliency prediction, we follow Jiang et al. (2019) in our experimental setup. DCSNet is trained from scratch over the training set of SALICON Jiang et al. (2015), following Jiang et al. (2019). We test our trained model on 3 widely used image saliency datasets, i.e., MIT1003Cornia et al. (2016), CAT2000Borji & Itti (2015), and DUTYang et al. (2013) and compare using 4 metrics: area under the curve (AUC), normalized scanpath saliency (NSS), CC and KL divergence.

In Table 11, we present a comparison with only published complex-valued saliency prediction methodJiang et al. (2019). As shown in the table, our method performs best overall.

Table 10: Comparison with FCCNYadav & Jerripothula (2023) across diffrent number of parameters.

| ImageNet | FCCN | | | DCSNet | | |
|---|---|---|---|---|---|---|
| | Resnet18 | ResNet50 | ResNet152 | Encoder | | |
| Param (M) | 11 | 26 | 60 | 5 | 15 | 17 |
| Acc (%) | 73.41 | 76.26 | 77.27 | 71.24 | 76.07 | **78.83** |

# F    QUALITATIVE RESULTS

In addition to all quantitative evaluations, we also validate our results qualitatively. We present these results in Fig. 8. The figure verifies our empirical claims when comparing with the complex-valued method as well as the real-valued method.

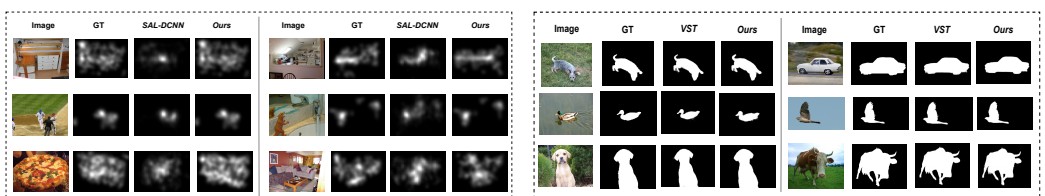

(a) Saliency map compared with SAL-DCNNJiang et al. (2019)
(b) Salient Objects compared with VSTLiu et al. (2021)

Figure 8: Qualitative results of our proposed CSNet compared with other methods.

Table 11: Comparison with existing complex-valued method SAL-DCNN Jiang et al. (2019)(light gray). Following Jiang et al. (2019), we take $201$, $400$, and $1,035$ test images for MIT1003, CAT2000 and DUT respectively. Our DCSNet obtains best results across all three datasets.

| | MIT1003Cornia et al. (2016) | | | | CAT2000Borji & Itti (2015) | | | | DUTYang et al. (2013) | | | |
|---|---|---|---|---|---|---|---|---|---|---|---|---|
| | AUC | NSS | CC | KL | AUC | NSS | CC | KL | AUC | NSS | CC | KL |
| SALICONJiang et al. (2015) | 0.82 | 1.28 | 0.42 | 1.61 | 0.77 | 0.99 | 0.39 | 1.17 | 0.85 | 2.27 | 0.48 | 1.24 |
| DVAWang & Shen (2018) | 0.86 | 2,19 | 0.66 | 0.87 | 0.81 | 1.50 | 0.56 | 0.84 | 0.91 | 3.11 | 0.67 | 0.88 |
| SALGANPan et al. (2017) | 0.87 | 2.05 | 0.65 | 0.96 | 0.81 | 1.47 | 0.56 | 0.97 | 0.91 | 2.80 | 0.68 | 0.90 |
| ML-NetCornia et al. (2016) | 0.84 | 2.01 | 0.61 | 1.01 | 0.79 | 1.37 | 0.51 | 0.99 | 0.88 | 2.87 | 0.61 | 1.15 |
| SAMCornia et al. (2018) | 0.87 | 2.19 | 0.61 | 1.30 | 0.84 | 1.74 | 0.63 | 1.12 | 0.91 | 2.96 | 0.67 | 1.07 |
| BMSZhang & Sclaroff (2016) | 0.77 | 1.15 | 0.37 | 1.43 | 0.78 | 1.20 | 0.46 | 1.07 | 0.83 | 1.76 | 0.42 | 1.40 |
| PQFTGuo & Zhang (2010) | 0.70 | 0.78 | 0.25 | 1.67 | 0.75 | 0.98 | 0.37 | 1.18 | 0.77 | 1.26 | 0.33 | 1.53 |
| SRHou & Zhang (2007) | 0.70 | 0.80 | 0.25 | 1.69 | 0.72 | 0.87 | 0.32 | 6.05 | 0.67 | 0.70 | 0.15 | 3.54 |
| Sal-DCNNJiang et al. (2019) | 0.87 | 2.10 | 0.62 | 0.89 | 0.86 | 2.03 | 0.79 | 0.63 | 0.92 | 3.07 | 0.76 | 0.55 |
| Sal-DCNN-PPJiang et al. (2019) | 0.86 | 1.98 | 0.61 | 0.93 | 0.86 | 2.00 | 0.77 | 0.65 | 0.92 | 3.06 | 0.75 | 0.57 |
| Sal-DCNN-PJiang et al. (2019) | 0.86 | 1.97 | 0.60 | 0.98 | 0.86 | 1.99 | 0.76 | 0.74 | 0.92 | 3.05 | 0.75 | 0.60 |
| Sal-DenseNetJiang et al. (2019) | 0.85 | 1.95 | 0.59 | 1.04 | 0.86 | 1.95 | 0.74 | 0.97 | 0.91 | 3.03 | 0.74 | 0.63 |
| **DCSNet (ours)** | **0.93** | **2.35** | **0.71** | **0.82** | **0.89** | **2.26** | **0.83** | **0.57** | **0.95** | **3.23** | **0.81** | **0.52** |

