# OpenReview forum: "Deep Complex Spatio-Spectral Networks with Complex Visual Inputs"
_ICLR.cc/2025/Conference — Submitted to ICLR 2025_

### Official Review · Reviewer_B2bV · 2024-10-29

**Soundness:** 4
**Presentation:** 4
**Contribution:** 4
**Rating:** 8
**Confidence:** 2

**Summary:**

This study investigates a novel complex-valued deep CNN designed for foreground extraction. It proposes a new method for encoding RGB images as complex values, an end-to-end token-based architecture that maintains the complex representation throughout, and an improved training pipeline. The authors demonstrate superior performance on a variety of complex-valued image benchmarks.

**Strengths:**

I am unfamiliar with the literature on complex-valued neural networks, and defer to the opinion of more reviewers. As a neophyte to this field, I found the manuscript overall to be very readable, well-written, interesting, and convincing. The novel encoding, architecture, and training pipeline seem to work well, and produce a very capable model for handling complex-valued inputs.

**Weaknesses:**

This may be my ascribed to my naivety for the field, but I'm unsure how to interpret the benchmark results. While DCSNet certainly seems to outperform other complex-valued neural networks, the margin of victory is often in the range of 1-5 percent. It is difficult to tell whether this represents a fundamental advance, or a marginal improvement. Further, Table 2 seems to indicate that complex-valued neural networks in general frequently fail to outperform their real-valued counterparts. How should these results be interpreted with respect to the broader viability of complex-valued neural networks?

**Questions:**

See weaknesses above.

---

> ### Author Response · Authors · 2024-12-02
> **Author response to reviewer B2bV**
>
> Thanks a lot for your encouraging comments and the rating.
>
> ## Weakness: Interpretation of Benchmark Results
>
> **Comparison with Other Complex-Valued Neural Networks:** To address your concern, we computed the average of the best accuracies achieved by existing complex-valued neural networks and compared them with our results. The averages were 87.55% for existing methods and 90.39% for our approach. This represents a significant improvement of 2.84%. Importantly, we also surpassed the 90% threshold, underscoring the potential of complex-valued neural networks in the computer vision domain and highlighting the significance of our contributions.
>
> **Comparison with Real-Valued Counterparts:** We would like to clarify that the real-valued networks mentioned in the paper as competitors are not exact real-valued counterparts due to differences in network size, network-type, pre-training methods, and other factors. Following Reviewer BUvX's suggestion, we have now conducted a direct comparison with actual real-valued counterparts, ensuring parity in configuration. As shown in the tables below, our approach demonstrates superior performance compared to these actual real-valued counterparts.
>
>
> For Salient Object Detection:
> |   Dataset |     |    DUTS    |           |         |  |  HKU-IS         |           |         | |  ECSSD         |           |         |
> |---------------|---------|--------|-----------|---------|---------|--------|-----------|---------|---------|--------|-----------|---------|
> |    | $S_m \uparrow$ |  $maxF\uparrow$ |  $E_{\xi}^{max}\uparrow$ |  $MAE\downarrow$ | $S_m \uparrow$ |  $maxF\uparrow$ |  $E_{\xi}^{max}\uparrow$ |  $MAE\downarrow$ |$S_m \uparrow$ |  $maxF\uparrow$ |  $E_{\xi}^{max}\uparrow$ |  $MAE\downarrow$ |
> | Real-DCSNet | 0.864 | 0.832 | 0.912 | 0.043 | 0.908 | 0.920 | 0.948 | 0.041 | 0.907 | 0.912 | 0.956 | 0.034 |
> | DCSNet | **0.894** | **0.874** | **0.941** | **0.039** | **0.927** | **0.945** | **0.960** | **0.034** | **0.917** | **0.924** | **0.967** | **0.029** |
>
>
>
>
> | Dataset |  | PASCAL-S    |           |         |    | DUT-O       |           |         |
> |---------|---------|:-----------:|-----------|---------|---------|--------|-----------|---------|
> |  |$S_m \uparrow$ |  $maxF\uparrow$ |  $E_{\xi}^{max}\uparrow$ |  $MAE\downarrow$ |$S_m \uparrow$ |  $maxF\uparrow$ |  $E_{\xi}^{max}\uparrow$ |  $MAE\downarrow$ |
> |Real-DCSNet| 0.847 | 0.841 | 0.896 | 0.069 | 0.817 | 0.747 | 0.851 | 0.062 |
> |DCSNet| **0.866** | **0.850** | **0.903** | **0.062** | **0.839** | **0.776** | **0.880** | **0.056** |
>
> &nbsp;
>
> For Defocus Blur Detection:
>
> | Method | DUT | | CUHK | |
> |------|-----|-----|-----|-----|
> |        | $\mathcal{F}_{\beta} \uparrow$ | MAE$\downarrow$ | $\mathcal{F}_{\beta} \uparrow$ | MAE$\downarrow$ |
> | Real-DCSNet | 0.841| 0.121 | 0. 899 | 0.064 |
> | DCSNet | **0.894**| **0.058** | **0.907** | **0.045**|
>
> &nbsp;
>
> For Shadow Detection:
> | Method | ISTD | SBU |
> |-----|-------|-------|
> |  | BER $\downarrow$ | BER $\downarrow$ |
> | Real-DCSNet | 1.83 | 3.45 |
> | DCSNet | **1.49** | **3.05** |
>
> &nbsp;
>
> For InSAR Foreground Extraction:
>
> | Method | mIoU $\uparrow$|
> |-------|---------|
> |Real-DCSNet | 0.85 |
> |DCSNet| **0.89** |
>
>
> We have also updated our Table 2 with the average results (shown below) obtained over the 5 datasets used. While our method achieves the best performance in terms of the maxF metric, it consistently ranks at least second-best across all metrics—a distinction no other method in the comparison achieves.
>
> | Method | $S_m \uparrow$ | $maxF\uparrow$ | $E_{\xi}^{max} \uparrow$ | $MAE\downarrow$ |
> |--------|--------|--------|--------|--------|
> |PiCANet     | 0.871 | 0.854 | 0.913 | 0.046 |
> |BASNet      | 0.872 | 0.856 | 0.909 | 0.051 |
> |PoolNet     | 0.879 | 0.864 | 0.914 | 0.049 |
> |EGNet-R     | 0.886 | 0.868 | 0.918 | 0.048 |
> |MINet-R     | 0.885 | 0.868 | 0.919 | 0.046 |
> |LDF-R       | 0.892 | 0.876 | 0.921 | `0.043` |
> |CSF-R2      | 0.892 | 0.874 | 0.911 | 0.049 |
> |GateNet-R   | 0.888 | 0.874 | 0.922 | 0.047 |
> |VST         | `0.904` | **0.894** | `0.932` | 0.045 |
> |FCCN        | 0.824 | 0.813 | 0.878 | 0.078 |
> |SCVUNet     | 0.831 | 0.827 | 0.893 | 0.069 |
> |DCSNet (Ours) | **0.893** | `0.895` | **0.930** | **0.044**|
>
> *Note: `Red` indicates best result, and **bold** indicates second best result.*

---

### Official Review · Reviewer_QqAh · 2024-10-30

**Soundness:** 3
**Presentation:** 3
**Contribution:** 2
**Rating:** 6
**Confidence:** 2

**Summary:**

This paper introduces the Deep Complex Patio-Spectral Network (DCSNet), a fully complex-valued, token-based neural network developed for end-to-end foreground extraction and adaptable for image classification. Extensive experiments show that DCSNet surpasses current complex-valued approaches across various tasks with real and complex-valued data, achieving results on par with leading real-valued models.

**Strengths:**

This paper presents a novel complex-valued neural network, the Deep Complex Patio-Spectral Network (DCSNet), a fully complex-valued, token-based, end-to-end architecture designed for foreground extraction and adaptable to image classification tasks. Extensive experiments demonstrate that DCSNet outperforms existing complex-valued methods across diverse tasks involving both real and complex-valued data, achieving competitive results relative to state-of-the-art real-valued models. The paper is well-written, concise, and easy to follow.

**Weaknesses:**

Novelty: The novelty of this paper appears questionable. The authors claim that they propose the first token-based complex-valued network that maintains complex-valued information throughout. However, the fully Complex-valued Convolutional Network (FCCN) [1] also processes complex-valued data through the entire model. Could the authors clarify any differences between these two models? What advantages does DCSNet offer over FCCN?

[1] Saurabh Yadav; Koteswar Rao Jerripothula, FCCNs: Fully Complex-valued Convolutional Networks using Complex-valued Color Model and Loss Function. ICCV 2023.

Complex-valued Image Generation (R2C Method): The authors propose an R2C method for generating complex-valued images from real-valued images, presenting it as a novel complex-valued color transformation. However, methods such as quaternion representation, complex logarithmic transformation, and the Hilbert transform are well-established for generating complex-valued images. Could the authors specify the advantages of the R2C method over these alternatives?

DCSNet Architecture:
1. Fourier filters replace self-attention in DCSNet to retain information within the complex domain while preserving global context. How do Fourier filters achieve global information retention in this context, and why were they chosen?
2. The paper briefly mentions dense tokens for image embedding but doesn’t fully explain their purpose. Are these tokens meant to capture pixel-level details (dense information) of the image? If so, why not use high-resolution Fourier filters as localized filters to capture this information directly?
3. If large Fourier filters serve as global filters in the frequency domain while dense tokens capture image details, could a bank of wavelet filters offer a more effective solution? Wavelet filters with multiple resolutions could extract both global (large scale) and local (small scale) image features.

Resolution Tokens: The paper mentions multiple resolution tokens \T_{i} for i \in {0,1,2,3}. What was the reasoning for using exactly four resolutions? Would using more or fewer resolutions impact the results?

Table 7 Clarification: In Table 7, there is a term \calL_{isal}. Is this a typo? Please clarify its meaning if not.

**Questions:**

Please see the weakness.

---

> ### Author Response · Authors · 2024-11-27
> **Author response to reviewer QqAh (part 1)**
>
> ## Weakness 1: DCSNet vs FCCN
>
> We thank the reviewer for the comment and appreciate the opportunity to clarify the differences between our proposed DCSNet and the FCCN. While both models maintain complex-valued information throughout, their design, application, and computational properties significantly differ. Below, we highlight these distinctions:
>
> **Application Domain**: FCCNs were primarily designed for image classification, whereas our DCSNets are specifically tailored for binary image segmentation. This fundamental difference in objectives influences both architectural and operational choices.
>
> **Processing Mechanism**: FCCNs utilize a sliding-window concept during convolution, while DCSNets leverage a tokenization approach inspired by transformer architecture, enabling efficient feature extraction and processing.
>
> **Complexity**: FCCNs exhibit quadratic complexity due to the convolution operation involved, whereas DCSNets achieve log-linear complexity due to the incorporation of the Fast Fourier Transform (FFT), making DCSNets computationally more efficient.
>
> **Operational Domains**: FCCNs operate entirely in the spatial domain, while DCSNets uniquely combine operations in both the spatial and frequency domains, facilitating a richer representation of complex-valued data.
>
> **Computational Efficiency and Performance**: DCSNet outperforms FCCNs both in terms of computational requirements and performance, as shown in the comparison table below:
>
> |  | FCCN (Resnet152) | DCSNet-encoder|
> |---|------|-------|
> |gFLOPS $\downarrow$ | 28 | **6** |
> |GPU memory (MB) $\downarrow$| 330 | **132**|
> | Time (ms) $\downarrow$| 61 | **13** |
> | Params (M) $\downarrow$| 59.8 |**16.9**|
> | Top-1 Accuracy $\uparrow$| 77.3 |**78.8**|
>
>
> These results demonstrate that our DCSNet-encoder not only achieves better accuracy but also significantly reduces computational costs, making it a superior choice, especially for resource-constrained environments.
> We hope this clarification addresses the reviewer’s concerns and illustrates the unique contributions and advantages of our DCSNet over FCCN. Thank you for your feedback, and we are happy to provide additional details if needed.

---

> > ### Comment · Reviewer_QqAh · 2024-12-01
> >
> > Thank you for explaining the differences between DCSNet and FCCN. However, I still do not fully understand the advantages of the proposed Complex-Valued Image Generation (R2C) method compared to other established alternatives, such as quaternion representation, complex logarithmic transformation, and the Hilbert transform. Perhaps I am missing a critical point. Could you clarify what makes the R2C method novel or unique?
> >
> > Additionally, regarding Question 3 on the DCSNet architecture, I did not find the response sufficiently clear or detailed to address my concerns. This may stem from my limited expertise in this area. As such, I have decided to retain my original score, albeit with lower confidence.

---

> ### Author Response · Authors · 2024-11-30
> **Author response to reviewer QqAh (part 2)**
>
> ## Weakness 4 & 5: Resolution tokens & Table 7 clarification
>
>
> **Resolution tokens**
>
> Using exactly four resolutions was an architectural design choice inspired by prior works [1], [2], and [3], which effectively leverage four distinct resolutions for multi-scale feature representation.
>
> To address your concern, we conducted additional experiments using three and five resolutions, and we present the results in the tables below. The findings show that using five resolutions improves some metrics (e.g., maxF and MAE), while slightly lowering others (e.g., $S_m$ and $E_{\xi}^{max}$). Conversely, using three resolutions resulted in a slight performance drop across all datasets and metrics compared to the four-resolution setting. This suggests that four resolutions provide a balanced trade-off across all evaluation criteria.
> Experiment with 3,4 and 5 resolutions:
>
> |    Dataset           |     |  DUTS   |           |         |   | ECSSD    |           |         | |  HKU-IS      |           |         |
> |---------------|---------|--------|-----------|---------|---------|--------|-----------|---------|---------|--------|-----------|---------|
> |      | $S_m \uparrow$ |  maxF$\uparrow$ |  $E_{\xi}^{max}\uparrow$ |  MAE$\downarrow$ | $S_m \uparrow$ |  maxF$\uparrow$ |  $E_{\xi}^{max}\uparrow$ |  MAE$\downarrow$ | $S_m \uparrow$ |  maxF$\uparrow$ |  $E_{\xi}^{max}\uparrow$ |  MAE$\downarrow$ |
> | 3 resolutions | 0.881 | 0.857 | 0.929 | 0.041 | 0.919 | 0.920 | 0.951 | 0.034 | 0.903 | 0.908 | 0.955 | 0.031 |
> | 4 resolutions | **0.894** | 0.874 | **0.941** | **0.039** | **0.927** | 0.945 | **0.960** | **0.034** | **0.917** | **0.924** | **0.967** | **0.029** |
> | 5 resolutions | 0.893 | **0.876** | 0.939 | **0.039** | **0.927** | **0.946** | 0.957 | 0.035 | 0.913 | **0.924** | 0.966 | **0.029** |
>
>
>
> | Dataset |  |  PASCAL-S   |           |         |      |    DUT-O    |           |         |
> |---------|---------|--------|-----------|---------|---------|--------|-----------|---------|
> | | $S_m \uparrow$ |  maxF$\uparrow$ |  $E_{\xi}^{max}\uparrow$ |  MAE$\downarrow$ | $S_m \uparrow$ |  maxF$\uparrow$ |  $E_{\xi}^{max}\uparrow$ |  MAE$\downarrow$ |
> | 3 resolutions |  0.851 | 0.833 | 0.897 | 0.064 | 0.819 | 0.756 | 0.848 | 0.058 |
> | 4 resolutions | **0.866** | 0.850 | **0.903** | **0.062** | **0.839** | 0.776 | **0.880** | **0.056** |
> | 5 resolutions |  0.862 | **0.852** | **0.903** | 0.064 | 0.835 | **0.779** | 0.879 | **0.056** |
>
>
> While using five resolutions can provide slight improvements in specific metrics, the four-resolution setup balances performance across datasets and metrics and aligns with prior research. We appreciate your suggestion, as it allowed us to validate and demonstrate the robustness of our approach under different architectural configurations.
>
> [1] Huang, Z., Dai, H., Xiang, T. Z., Wang, S., Chen, H. X., Qin, J., & Xiong, H. (2023). Feature shrinkage pyramid for camouflaged object detection with transformers. In Proceedings of the IEEE/CVF conference on computer vision and pattern recognition (pp. 5557-5566).
>
> [2] Liu, N., Zhang, N., Wan, K., Shao, L., & Han, J. (2021). Visual saliency transformer. In Proceedings of the IEEE/CVF international conference on computer vision (pp. 4722-4732).
>
> [3] Wang, W., Xie, E., Li, X., Fan, D. P., Song, K., Liang, D., ... & Shao, L. (2021). Pyramid vision transformer: A versatile backbone for dense prediction without convolutions. In Proceedings of the IEEE/CVF international conference on computer vision (pp. 568-578).
>
> &nbsp;
> &nbsp;
>
>
> **Table 7 clarification**
>
> Thank you for your careful observation. The term was indeed a typo. We have rectified this to $\mathcal{L}_{idense}$ in the revised manuscript.

---

> ### Author Response · Authors · 2024-12-01
> **Author response to reviewer QqAh (part 3)**
>
> ## Weakness 2: R2C vs existing alternatives
>
> **Apologies for the delayed responses; the lead author was not doing well. We will respond to the question on DCSNet architecture as well in few hours.**
>
> We appreciate the comment on comparison with well-established methods of complex-valued image generation. While techniques such as quaternion representation, complex logarithmic transformation, and the Hilbert transform are indeed powerful tools, our proposed R2C method offers some key advantages, which we discuss below.
>
> **Intuitive Mapping of Color to Complex Plane:** Our R2C method provides a more intuitive and geometrically grounded mapping of colors to the complex domain. By identifying two distinct Argand planes in the RGB color space itself, our method captures both the color relationship with respect to the grayscale (grayline) and the spatial positioning of the color within the RGB cube, resulting in a complex-valued color model called iRGB. We hardly have any complex-valued color models except iHSV (proposed in FCCNs paper [1]), and our experiments have shown better performance of our iRGB over iHSV (see Table 7).
>
> **Richer & Dual Complex Representations:** This approach enables us to represent each color as two complex numbers, which provides a richer representation than typical methods which only focus on one color channel (e.g., Hilbert transform) at a time or a single transformation (e.g., using quaternions, where a 4D complex number [$0 + iR + jG + kB$] is employed without considering the inherent color space geometry). The two complex numbers derived from the Argand planes allow our color model (iRGB) to preserve more nuanced information about color differences and the relationship between color and luminance. Through this dual complex representation, our method captures the perceptual relationship between luminance and chrominance in a way that is not explicitly addressed by the quaternion or Hilbert transform approaches. This makes our R2C method applicable to a broader range of image processing algorithms.
>
> **Computational Efficiency:** The R2C method transforms real-valued images to complex-valued images by processing each pixel independently, resulting in a computational complexity of $O(n)$ and making it highly parallelizable. In contrast, methods like the Hilbert transform have higher computational costs, at least $O(n\log n)$. This efficiency makes R2C well-suited for large-scale image processing tasks.
>
> **Flexibility in Applications:** While methods like quaternion representation or complex logarithmic transformations typically focus on specific kinds of data (e.g., 3D rotations or magnitude-phase representations), our method is more flexible in mapping any color into a pair of complex numbers, making it a general framework for complex-valued image generation.
>
> In summary, while quaternion representations and other well-established transformations have their place in the broader context of complex-valued image processing, the R2C method offers a unique advantage by providing an intuitive, dual-complex, efficient representation of color that better captures the geometry of the RGB color space, while leading to wider application domains and applications in various image processing algorithms. We believe this contribution significantly advances the ability to generate complex-valued images from real-valued ones.
>
> We hope this clarifies the distinction between our method and existing alternatives and demonstrates its utility in the context of complex-valued image generation. We will get back to you on our architecture soon.
>
> [1] Saurabh Yadav; Koteswar Rao Jerripothula, FCCNs: Fully Complex-valued Convolutional Networks using Complex-valued Color Model and Loss Function. ICCV 2023.

---

> ### Author Response · Authors · 2024-12-01
> **Author response to reviewer QqAh (part 4)**
>
> ## Weakness 3 [part 1 of 2]: DCSNet Architecture
>
> 1. In DCSNet, we maintain a complex-valued representation throughout the network, which requires careful design to balance computational efficiency and representational power. Below, we explain why Fourier filters were chosen and how they act globally:
>
>     Fourier filters are inherently global because their operation spans all frequencies simultaneously through element-wise multiplication in Fourier domain, which inherently incorporates information across the entire image due to the global nature of the Fourier transform.
>
>     Using complex-valued self-attention to handle the complex-valued inputs in DCSNet would require additional multiplication operations, which significantly increases computational complexity. Following [1], we explored alternatives and presented a comparison in Appendix D. Fourier filters offer a computationally efficient solution ([2]), as they leverage the properties of the Fourier domain to handle global information with lower computational overhead compared to self-attention mechanisms.
>
>
>
> &nbsp;
> &nbsp;
>
> 2. We appreciate the reviewer’s observation regarding the use of dense tokens and their purpose in our method. Below, we clarify the role of dense tokens and the rationale for their inclusion in our design:
>
>
>      Dense tokens are additional learnable tokens specifically designed for dense prediction tasks, such as predicting a binary mask at different resolutions. Rather than altering or replacing the image patch tokens, dense tokens interact with these patch tokens during the forward pass to learn task-specific, image-dependent embeddings. This mechanism is inspired by transformer-based methods ([3], [4]) that utilize such tokens for specific downstream tasks.
>     By maintaining the dense tokens as separate embeddings, we ensure that the image information remains intact while enabling the network to learn representations tailored for dense output predictions.
>
>
>     We agree that high-resolution Fourier filters could potentially capture localized features, and that is why we have incorporated multi-scale architecture, with initial layers of encoder and final layers of decoder having high-resolution Fourier filters. We would like to clarify that We called Forier filters global only due to the kind of operation they perform. Fourier filters can capture local information as well.
>
>     For example, Even a small filter of size 3×3 in the spatial domain, capable of capturing local features, must first be zero-padded to match the image dimensions before being transformed into the Fourier domain to produce a filter of the same size for multiplication. We hypothesize that the network can learn to construct such Fourier filters that, when interpreted in the spatial domain, effectively have a smaller receptive field, enabling the extraction of local features when needed.
>     On the other hand, dense tokens are explicitly trained to learn spatially-aware embeddings required for downstream dense prediction tasks, using features (be it global or local) learned using Forier filters.
>
>     In summary, we would like to clarify that Fourier filters are used for feature learning, and dense tokens help in executing downstream tasks.
>
>     We conducted an ablation study (Table 6 in the manuscript) to evaluate the impact of removing dense tokens. This study demonstrated a performance degradation when dense tokens were not used, highlighting their importance for accurate dense prediction.
>
> &nbsp;
> &nbsp;
>
>
> [1] Eilers, F., & Jiang, X. (2023, June). Building Blocks for a Complex-Valued Transformer Architecture. In ICASSP 2023-2023 IEEE International Conference on Acoustics, Speech and Signal Processing (ICASSP)
>
> [2] Rao, Y., Zhao, W., Zhu, Z., Lu, J., & Zhou, J. (2021). Global filter networks for image classification. Advances in neural information processing systems
>
> [3] Dosovitskiy, A., Beyer, L., Kolesnikov, A., Weissenborn, D., Zhai, X., Unterthiner, T., Dehghani, M., Minderer, M., Heigold, G., Gelly, S., Uszkoreit, J., & Houlsby, N. (2021). An Image is Worth 16x16 Words: Transformers for Image Recognition at Scale. ICLR.
>
> [4] Liu, N., Zhang, N., Wan, K., Shao, L., & Han, J. (2021). Visual saliency transformer. In Proceedings of the IEEE/CVF international conference on computer vision (pp. 4722-4732).

---

> ### Author Response · Authors · 2024-12-01
> **Author response to reviewer QqAh (part 5)**
>
> ## Weakness 3 [part 2 of 2]: DCSNet Architecture
>
> 3. We thank the reviewer for suggesting the potential use of wavelet filters for multi-resolution analysis.
> While wavelet filters offer multi-resolution analysis by localizing both spatial and frequency components, incorporating a bank of wavelet filters in the current framework would come with its own set of challenges. Wavelet filters require multiple scales and orientations, which can significantly increase the computational and memory overhead, especially when working with high-dimensional image data. While Wavelet filters inherently capture local features, they need to be predefined and fixed. We would need to define learnable wavelet filters in complex-valued neural networks, which is definitely interesting and can be explored. However, as we have already clarified above, our Fourier filters are taking care of both global and local feature extraction; we need not make the suggested replacement. Moreover, we believe the added advantage offered by the wavelet transform is already being offered through our multi-scale architecture.
>
> We appreciate the reviewer for suggesting some great ideas worth exploring in the context of complex-valued networks.
>
> &nbsp;
> &nbsp;

---

> > ### Comment · Reviewer_QqAh · 2024-12-02
> >
> > Thank you for the clarification. The authors have addressed most of my concerns. However, I believe that multi-scale wavelet filters could offer potential improvements over Fourier filters, which could be a worthwhile direction for future exploration. I am raising my score to 6, albeit with reduced confidence.

---

> > > ### Author Response · Authors · 2024-12-03
> > > **Discussion Follow-up (for Reviewer QqAh)**
> > >
> > > Dear Reviewer QqAh,
> > >
> > > Thank you very much for taking the time to carefully reconsider our paper and for your updated evaluation. We truly appreciate your thoughtful feedback and the engaging discussion we had.
> > >
> > > Your insights and comments played a crucial role in helping us refine and present our work more effectively. We are particularly grateful for your suggestion to explore wavelet filters; it is an excellent idea that we will certainly pursue in our future work.
> > >
> > > It was a pleasure to have you as a reviewer, and we deeply value your contributions to the review process. Thank you once again for your time and support.
> > >
> > > &nbsp;
> > >
> > > Warm regards,
> > >
> > > Authors of #1276

---

### Official Review · Reviewer_F89z · 2024-11-02

**Soundness:** 3
**Presentation:** 2
**Contribution:** 1
**Rating:** 3
**Confidence:** 4

**Summary:**

The work proposes a complex-value deep neural network for computing vision tasks. First, the authors propose an inevitable real-to-complex transformation. Then, the work proposes an architecture comprising spectral convolution and a complex T2T module.
The authors evaluated their model on image classification, smooth object detection, and defocus blur detection.

**Strengths:**

The work proposes a novel invertible real-to-complex conversion for RGB images in complex-valued neural networks. The procedures are clearly stated using pseudocode and figures.

**Weaknesses:**

1. The authors seem to miss the seminal work on complex values networks and didn’t compare/discuss with the techniques discussed in [1]
2. The work is directed at using complex-valued networks for real-valued images. The majority of the paper involves devising an invertible conversion from real to complex representation. However, the paper fails to demonstrate its utility. For example, for classification on ImageNets, the authors did not consider state-of-the-art models, such as Vit, Swin-v2, etc., which achieve above 90% accuracy.

3. The paper does not discuss the motivation of the specific real to complex conversion. There are many invertible conversions between real and complex.

[1] DEEP COMPLEX NETWORKS

**Questions:**

1. Line 317: Citation link broken for T2T-ViTYuan et al. (2021)

2. Line 269: Do you use complex-valued  “ normalization” as discussed in [1]

3. How well does the model perform if we consider trivial real to complex conversion that considers the real numbers as complex numbers with $0$ imaginary part? This is also a very crucial ablation that the authors should perform.

4. Does using spectral convolution make it challenging to capture local features as it performs global convolution?


[1] DEEP COMPLEX NETWORKS

---

> ### Author Response · Authors · 2024-11-24
> **Author response to reviewer F89z (part 1)**
>
> ## Weakness 1 & Question 2: Discussion and comparison with seminal work
>
> Thank you for pointing out that we missed discussing DCN [1], which is indeed a seminal work in this domain and should have been included in our discussion. DCN introduced foundational concepts for complex-valued neural networks, including complex-valued convolution, activation functions, and normalization. However, we note that DCN essentially operates in the real domain for both input and output. Specifically, it assumes images as real-valued input and then derives the imaginary part from that very real-valued input (refer to our response to reviewer BUvX [in part 4] for further clarification). Additionally, the complex-valued nature of DCN is restricted to the convolutional base. The real and imaginary components are concatenated and passed to real-valued fully connected layers. Furthermore, its evaluation is limited to image classification tasks (as far as the computer vision domain is concerned), primarily on small-scale datasets.
>
> In contrast, our work addresses the challenge of building a fully complex-valued pipeline, where all components—from input to output, including the loss function—are in the complex domain. While FCCN [2] also addresses these by taking a fully convolutional approach, proposing the iHSV color space and a complex-valued loss function, its scope is again limited to image classification. Our work moves beyond this by adopting a token-based architecture inspired by transformers, utilizing Fourier filters as fundamental building blocks in both the encoder and decoder. Crucially, we focus on binary segmentation problems, which is more challenging.
>
> As we needed a backbone trained on a large dataset for building our binary segmentation network, we trained a robust complex-valued encoder on ImageNet, achieving the best results to date for any complex-valued network. The comparative results are summarized in the table below:
>
> | DCN [1] | FCCN [2]  | DCSNet (Ours) |
> |------|--------|--------|
> |  72.6 |  77.3 | **78.8**|
>
> Note that the ImageNet results for DCN [1] were taken from FCCN [2], which we missed in the original version. This oversight has been corrected now in the paper as well. Finally, we would like to confirm that we used complex-valued normalization, as described in DCN [1].
>
> References:
>
> [1] C. Trabelsi, O. Bilaniuk, Dmitriy Serdyuk, Sandeep Subramanian, J. F. Santos, Soroush Mehri, Negar Rostamzadeh, Yoshua Bengio, C. Pal, "Deep Complex Networks," ICLR 2018.
>
> [2] Saurabh Yadav, Koteswar Rao Jerripothula, "FCCNs: Fully Complex-valued Convolutional Networks using Complex-valued Color Model and Loss Function," ICCV 2023.

---

> ### Author Response · Authors · 2024-11-24
> **Author response to reviewer F89z (part 2)**
>
> ## Weakness 2, Weakness 3 & Question 3: Real to Complex Conversion
>
> As Reviewer BUvX also pointed out, complex-valued neural networks can indeed outperform their real-valued counterparts, but we believe this largely depends on the nature of the input data. For instance, the seminal work DCN [1] achieved similar results for complex-valued and real-valued networks on image classification tasks. One plausible reason for this is the use of the RGB color space for input, which is inherently real-valued. In contrast, for audio tasks—where complex-valued data is naturally available—complex-valued networks performed better, as reported in [1]. This underscores the importance of the input domain: for complex-valued networks to fully exploit their potential, the input data should ideally reside in the complex-valued domain.
>
> This idea has been well-recognized in subsequent works, including FCCN [2], which proposed the iHSV color space to address this limitation. Complex-valued color models are critical for learning rich complex-valued representations, and we argue that the input data must also be complex-valued to facilitate this process effectively.
>
> It is important to note that this goes beyond transformations like Fourier transforms, which can indeed produce complex representations but do not allow for intuitive visualization like we experience via images with discernible corners, edges, and shapes. A complex-valued color model bridges this gap: it not only provides complex representations but also enables visualization. This visualization capability is particularly vital for binary segmentation tasks, where spatial and structural information is paramount.
>
>
> **Why a new complex-valued color model?**
>
> While iHSV has demonstrated success, ColorNet [3] shows that the choice of color model can significantly impact network accuracy. It also showed that using multiple color models simultaneously can further enhance accuracy while reducing the number of parameters. Therefore, having more complex-valued color models can certainly help research in the domain of complex-valued networks. Inspired by these insights, we explored RGB color model and searched for argand planes to develop a novel complex-valued color model, iRGB. Our experiments demonstrate that iRGB outperforms iHSV, as evidenced in Table 7 of our paper and the table below, which shows results on CIFAR10:
>
> | L+i0 & a+ib [4] &nbsp; | R+iG & G +iB [4] &nbsp;| Fourier Transform &nbsp;| iHSV [2] &nbsp; | RGB+i0 &nbsp; | iRGB (Ours) |
> |------|-----|------|-----|--------|--------|
> |  91.8 | 92.7 | 85.8 | 93.5 | 89.1 | **94.3**|
>
>
> **Comparison with other existing/trivial transformations**
>
>
> The literature includes various real-to-complex transformations, such as the two encodings proposed in [4]: (i) L+i0 & a+ib (using L*ab space) and (ii) R+iG & G+iB (using RGB space). We also tried the trivial transformation RGB+i0 you suggested in Q3. Previous works, such as DCN [1] (Section 3.7) and FCCN [2] (Table 6), have also empirically shown that having both real and imaginary parts contributes to better performance. Our results reaffirm this, highlighting the superior performance of iRGB. As demonstrated in the table above, our iRGB surpasses all other methods, including iHSV, in terms of accuracy.
>
>
> **Invertibility of the transformation**
>
>
> Regarding invertibility, it is crucial to ensure no information loss occurs during transformations. Otherwise, the network would operate on incomplete data. We have validated this property for iRGB and included additional experiments and an enhanced demo in the supplementary material, as suggested by Reviewer BUvX.
>
> &nbsp;
> &nbsp;
>
> [1] C. Trabelsi, O. Bilaniuk, Dmitriy Serdyuk, Sandeep Subramanian, J. F. Santos, Soroush Mehri, Negar Rostamzadeh, Yoshua Bengio, C. Pal, "Deep Complex Networks", ICLR 2018.
>
> [2] Saurabh Yadav; Koteswar Rao Jerripothula, FCCNs: Fully Complex-valued Convolutional Networks using Complex-valued Color Model and Loss Function. ICCV 2023.
>
> [3] Gowda, S.N., Yuan, C. (2019). ColorNet: Investigating the Importance of Color Spaces for Image Classification. In: Jawahar, C., Li, H., Mori, G., Schindler, K. (eds) Computer Vision – ACCV 2018.
>
> [4] Utkarsh Singhal, Yifei Xing, Stella X. Yu; Proceedings of the IEEE/CVF Conference on Computer Vision and Pattern Recognition (CVPR), 2022, pp. 681-690

---

> ### Author Response · Authors · 2024-11-24
> **Author response to reviewer F89z (part 3)**
>
> ## Weakness 2: Comparison with SOTA Image Classification Results
>
> As stated earlier, the primary objective of our work is to address **binary segmentation in the complex domain**. Achieving this requires a decent complex-valued backbone trained on a sufficiently large dataset, such as ImageNet. To this end, we trained our DCSNet encoder on ImageNet-1k, which demonstrated superior performance compared to the existing state-of-the-art complex-valued method for image classification, FCCN [2], as shown in Table 1 of our paper. Importantly, this improvement was achieved with fewer parameters.
>
> While we included some baselines from the real domain for reference in Table 1, our goal was never to achieve state-of-the-art (SOTA) image classification results overall. Rather, our aim was to obtain the **best binary segmentation results in the complex domain**, which we have consistently achieved (see Tables 2-5), as acknowledged by reviewer B2bV as well. Notably, our approach **excels on both real-valued and complex-valued datasets**, underscoring the versatility of our proposed method—a capability that real-valued networks cannot achieve without compromising the inherent complex-valued nature of the data. Additionally, while our primary focus was binary segmentation, in the process, our work also produced the best results in the complex domain for image classification tasks across both real-valued and complex-valued datasets (refer to Table 1).
>
>
>
> Moreover, modern state-of-the-art methods like ViT and Swin-V2 are designed and optimized specifically for image classification. These networks are trained on significantly larger datasets, such as JFT-3B and ImageNet-22k, and leverage over a billion parameters [1]. They are then fine-tuned on ImageNet-1k, achieving accuracies close to 90%. In contrast, our model was trained directly on ImageNet-1k, without leveraging such large-scale datasets or model sizes. Given this difference in objectives and resources, we feel it is not a fair comparison to benchmark our model against such methods, as image classification SOTA was never our goal. We just needed a good enough complex-valued backbone/encoder.
>
> Instead of allocating our limited computational resources toward achieving SOTA results on image classification—a task beyond our focus—we have dedicated our efforts toward building the **binary segmentation networks in the complex domain** that this work aims to deliver.
>
> References:
>
> [1] Liu, Z., Hu, H., Lin, Y., Yao, Z., Xie, Z., Wei, Y., ... & Guo, B. (2022). Swin transformer v2: Scaling up capacity and resolution. In Proceedings of the IEEE/CVF conference on computer vision and pattern recognition.
>
> [2] Saurabh Yadav; Koteswar Rao Jerripothula, FCCNs: Fully Complex-valued Convolutional Networks using Complex-valued Color Model and Loss Function. ICCV 2023.

---

> > ### Author Response · Authors · 2024-11-24
> > **Author response to reviewer F89z (part 4)**
> >
> > ## Question 4 & Question 1: Capturing local features & Citation link issue
> >
> > Fourier filters are inherently global, whether they are low-pass, high-pass, Butterworth, or notch filters, as their operation spans all frequencies simultaneously through multiplication in the Fourier domain. Even a small filter of size 3×3 in the spatial domain, capable of capturing local features, must first be zero-padded to match the image dimensions before being transformed into the Fourier domain to produce a filter of the same size for multiplication.
> >
> > Our hypothesis is that the network can learn to construct such Fourier filters that, when interpreted in the spatial domain, effectively have a smaller receptive field, enabling the extraction of local features when needed. Furthermore, our network employs multi-scale skip connections and multi-scale feature learning, facilitating feature learning at both global and local levels, as demonstrated in [1], [2], [3], and [4].
> >
> >
> > Thank you for pointing out the issue with the citation link; we have corrected it in the revised manuscript.
> >
> > References:
> >
> > [1] Nian Liu, Ni Zhang, Kaiyuan Wan, Ling Shao, and Junwei Han. Visual saliency transformer. IEEE/CVF International Conference on Computer Vision, ICCV 2021
> >
> >
> > [2] Wang, W., Xie, E., Li, X., Fan, D. P., Song, K., Liang, D., ... & Shao, L. (2021). Pyramid vision transformer: A versatile backbone for dense prediction without convolutions. In Proceedings of the IEEE/CVF international conference on computer vision.
> >
> >
> > [3] Zhang, W., Huang, Z., Luo, G., Chen, T., Wang, X., Liu, W., ... & Shen, C. (2022). Topformer: Token pyramid transformer for mobile semantic segmentation. In Proceedings of the IEEE/CVF Conference on Computer Vision and Pattern Recognition.
> >
> >
> > [4] Yan, X., Tang, H., Sun, S., Ma, H., Kong, D., & Xie, X. (2022). After-unet: Axial fusion transformer unet for medical image segmentation. In Proceedings of the IEEE/CVF winter conference on applications of computer vision.

---

> ### Comment · Reviewer_F89z · 2024-11-25
> **Response to Rebuttal**
>
> ## (Part 1)
> Thanks for the rebuttal.
>
> 1. "However, we note that DCN essentially operates in the real domain for both input and output." ----  In Section 4.2 of [1] they perform music spectrum prediction. I believe that is complex to complex mapping. Correct me if I am wrong.
>
> 2. " Additionally, the complex-valued nature of DCN is restricted to the convolutional base." --- Could you please clarify?
>
> 3.  "The real and imaginary components are concatenated and passed to real-valued fully connected layers." --- There is no difference in representing a complex number as $a+ib$ or $[a,b]$ as long as you model the operations of a complex number of, for example, see Eq 2 and Eq 6 of [1]. These are different from just concatenating the real and imaginary parts and passing them as real-valued input to the model.
>
>
>
>
>
> [1] C. Trabelsi, O. Bilaniuk, Dmitriy Serdyuk, Sandeep Subramanian, J. F. Santos, Soroush Mehri, Negar Rostamzadeh, Yoshua Bengio, C. Pal, "Deep Complex Networks," ICLR 2018.

---

> > ### Comment · Reviewer_F89z · 2024-11-25
> > **Response to Rebuttal**
> >
> > ## Part 2-4
> >
> > 1. Thanks for the ablation study for different real-to-complex conversions. When I asked for motivation, I sought a theoretical or intuitive explanation of "why the iRGB is better than others, such as $R+iG$ & $G +iB$".
> >
> > 2.  "best binary segmentation results in the complex domain" -- I believe that the authors meant the output segmentation mask is complex. I get that if the original input is complex-valued, it is better to use complex-valued models. However, why do we need the output mask to be $1$ and $i$ instead of $1$ and $0$?
> >
> > 3. I understand the authors' explanation that the filters can be learned to be local. However, we need to use very high-frequency components in constructing the spectral filter, which might be overfitting. Local constructions of special filters are discussed in [2]. (This point is just for discussion and improving the model.  It is not a weakness that the authors need to respond to.)
> >
> > [2] *Truly Scale-Equivariant Deep Nets with Fourier Layers*

---

> ### Author Response · Authors · 2024-12-02
> **Author Response to additional comments of Reviewer F89z (part 1)**
>
> ## Part 1
>
> Thanks for the additional comments. Here is our response.
>
> **1)** We would like to clarify that our statement was made specifically in the context of the computer vision task addressed by DCN [1], namely image recognition, where both inputs and outputs were indeed real-valued. Furthermore, as our paper focuses on *"DCSNets with complex visual inputs"* (as stated in the title), we have deliberately restricted all discussions, experiments, and rebuttals to the computer vision domain. The DCN paper itself acknowledged achieving only comparable results to real-valued counterparts in the computer vision task it undertook. To address this specific limitation, we developed the iRGB color model and a tailored loss function to effectively handle complex-valued outputs for the binary image segmentation task.
>
> Regarding the additional acoustic-related tasks tackled by DCN—namely automatic music transcription and speech spectrum prediction—we believe these are outside the scope of our paper, which is explicitly centered on computer vision. However, since you have now pointed out these audio-related tasks, we would like to provide the following clarification:
>
> In audio-related tasks, complex-valued data is naturally available. For the first task (Section 4.2), the outputs were real-valued. For the second task (Section 4.3), you are correct that it involves a complex-to-complex mapping, achieved in DCN by employing an entirely convolutional network. Since audio tasks are beyond our focus, we can only state that such mappings could also be tried using a complex-valued token-based architecture. While such exploration would indeed be interesting, it would distracting from the primary focus of this paper. Thank you for pointing this out; we will certainly consider pursuing this idea in our future work.
> &nbsp;
>
> **2)** As mentioned above, since DCN needed real-valued outputs for the image recognition task, the authors concatenated the real and imaginary outputs of the convolutional base to feed the subsequent real-valued fully connected layers. For reference, please see the official implementation of [1] on [github](https://github.com/ChihebTrabelsi/deep_complex_networks/blob/master/scripts/training.py) at line 250. It is clear from the code that real-valued fully connected layers are used. Therefore, the complex-valued nature of the network is limited to the convolutional base in the computer vision task undertaken by DCN.
>
> &nbsp;
>
> **3)** We have already clarified above that complex operations no longer occur in the fully connected layers of DCN (in image recognition task). Therefore, representing a complex number as $a+ib$ or $[a,b]$ in these fully connected layers does make a difference. You yourself mentioned, representing a complex number as $a+ib$ or $[a,b]$  are identical as long as complex operations are maintained. However, DCN performs complex-valued operations only in its convolutional base. In its fully connected layers, it performs real-valued operations on the real-valued vector $[a,b]$. On the other hand, we perform complex operations on $a+ib$ throughout our network.
>
> Regarding Eqns 2 & 6 in the DCN paper, these pertain to complex convolution and complex batch normalization, respectively, both of which are employed in the convolutional base, not in the fully connected layers.
>
> [1] C. Trabelsi, O. Bilaniuk, Dmitriy Serdyuk, Sandeep Subramanian, J. F. Santos, Soroush Mehri, Negar Rostamzadeh, Yoshua Bengio, C. Pal, "Deep Complex Networks," ICLR 2018.

---

> ### Author Response · Authors · 2024-12-02
> **Author Response to additional comments of Reviewer F89z (part 2)**
>
> ## (Part 2) Advantages of iRGB over others
> As Reviewer QqAh also raised the same point regarding iRGB's advantages over other representations, we have now listed them below:
>
> **Intuitive Mapping of Color to Complex Plane:** Our R2C method provides a more intuitive and geometrically grounded mapping of colors to the complex domain. By identifying two distinct Argand planes in the RGB color space itself, our method captures both the color relationship with respect to the grayscale (grayline) and the spatial positioning of the color within the RGB cube, resulting in a complex-valued color model called iRGB. We hardly have any complex-valued color models except iHSV (proposed in FCCNs paper [1]), and our experiments have shown better performance of our iRGB over iHSV.
>
> **Richer & Dual Complex Representations:** This approach enables us to represent each color as two complex numbers, which provides a richer representation than typical methods which only focus on one color channel (e.g., Hilbert transform) at a time or a single transformation (e.g., using quaternions, where a 4D complex number [$0 + iR + jG + kB$] is employed without considering the inherent color space geometry). The two complex numbers derived from the Argand planes allow our color model (iRGB) to preserve more nuanced information about color differences and the relationship between color and luminance. Through this dual complex representation, our method captures the perceptual relationship between luminance and chrominance in a way that is not explicitly addressed by the quaternion or Hilbert transform approaches. This makes our R2C method applicable to a broader range of image processing algorithms.
>
> **Computational Efficiency:** The R2C method transforms real-valued images to complex-valued images by processing each pixel independently, resulting in a computational complexity of $O(n)$ and making it highly parallelizable. In contrast, methods like the Hilbert transform have higher computational costs, at least $O(n\log n)$. This efficiency makes R2C well-suited for large-scale image processing tasks.
>
> **Flexibility in Applications:** While methods like quaternion representation or complex logarithmic transformations typically focus on specific kinds of data (e.g., 3D rotations or magnitude-phase representations), our method is more flexible in mapping any color into a pair of complex numbers, making it a general framework for complex-valued image generation.
>
> In summary, while quaternion representations and other well-established transformations have their place in the broader context of complex-valued image processing, the R2C method offers a unique advantage by providing an intuitive, dual-complex, efficient representation of color that better captures the geometry of the RGB color space, while leading to wider application domains and applications in various image processing algorithms. We believe this contribution significantly advances the ability to generate complex-valued images from real-valued ones.
>
>
> **Advantages of iRGB over simple [R+iG,G+iB] representation**
>
> The representation $[R+iG,G+iB]$ is a relatively naive approach and disregards key principles of color model design. Below, we highlight three specific advantages of our iRGB model:
>
> *Independence of Color Components*: In $[R+iG,G+iB]$, the imaginary part of the first component ($G$) is tied to the real part of the second component ($G$). This dependency inhibits the independent processing of color components. In contrast, our iRGB representation utilizes two fully independent components, $||v||e^{i\theta}$ and $||u||e^{i\phi}$, allowing greater flexibility and more effective color representation.
>
> *Unbiased Representation*: The $[R+iG,G+iB]$ approach inherently biases the green ($G$) color channel due to its repetition, potentially leading to suboptimal results. In comparison, iRGB avoids such biases, providing a balanced and unbiased representation across color channels.
>
> *Intuitive Meaning*: The $[R+iG,G+iB]$ representation lacks a clear and intuitive physical or conceptual interpretation. On the other hand, iRGB directly encodes both intensity and color information, offering a more meaningful and interpretable representation as detailed earlier.
>
> We hope this clarifies various advantages our iRGB offers over others.
>
> [1] Saurabh Yadav; Koteswar Rao Jerripothula, FCCNs: Fully Complex-valued Convolutional Networks using Complex-valued Color Model and Loss Function. ICCV 2023.

---

> ### Author Response · Authors · 2024-12-02
> **Author Response to additional comments of Reviewer F89z (part 3)**
>
> ## Part 3
>
> By "best binary segmentation results in the complex domain," we meant the best binary segmentation results achieved using complex-valued networks.
>
> There are two reasons why we chose the background to be represented by '$i$' instead of '$0$', while keeping the foreground as '$1$':
>
> *Complex-Valued Ground Truth Encoding*: To compare our complex output against the ground truth, a complex-valued encoding of the ground truth was required. We encoded the binary segmentation map $y$ as $y+i(1−y)$, which results in background pixels being represented by '$i$'.
>
> *Foreground and Background Independence*: Similar to the real and imaginary parts of a complex number, in the physical world, the foreground and background are conceptually independent of each other. This analogy inspired us to represent the foreground as the real part and the background as the imaginary part of the image. This convention appeared well-suited for addressing the binary image segmentation problem in the complex domain.
>
> &nbsp;
>
> **Local Feature Learning:**
>
> Thank you for pointing out reference [1], which proposed a localized Fourier layer for efficiently learning local features. The approach constrains the degrees of freedom of a Fourier filter to ensure that the corresponding spatial kernel is spatially localized. This is indeed an interesting idea, and we would be glad to explore it in our future work.
>
> [1] Rahman, M. A., & Yeh, R. A. (2023). Truly scale-equivariant deep nets with fourier layers. Advances in Neural Information Processing Systems, 36, 6092-6104.

---

> > ### Comment · Reviewer_F89z · 2024-12-03
> >
> > Thanks for the clarifications,
> > Thanks for the discussion on DCN. Even though the authors of DCN did not use complex, dense layer for the image classification tasks, they do provide the recipe for it [github](https://github.com/ChihebTrabelsi/deep_complex_networks/blob/master/complexnn/dense.py).
> >
> > `Regarding the real to complex conversion`
> >
> > Please add this discussion in the main text, as this 'real to convex' transformation is the paper's core contribution.
> >
> > ** Independence of Color Components ** Authors argued that in $ [R+iG, G+iB] $, the components are not independent. In the proposed method, aren't the magnitudes of the vectors $v$ and $u$ related? If they are related, then the complex components are not completely independent.
> >
> > `Binary Segmentation`
> >
> > "To compare our complex output against the ground truth..." Does this mean the proposed model is unsuitable for mapping complex input to real output? Otherwise, this choice seems forced.
> >
> > Also, I am unsure if I follow the reasoning of assigning $i$ instead of $0$ to the background to make it independent of the foreground.

---

> > > ### Author Response · Authors · 2024-12-03
> > > **Author Response to further comments of Reviewer F89z**
> > >
> > > `Github repository of DCN`
> > >
> > > Thanks for pointing it out. However, as we have already stated, they perform complex-to-complex mapping in the audio domain, so it must have been used there. To maintain focus, it will be nice if we restrict the discussion to the computer vision domain, as emphasized earlier.
> > >
> > > &nbsp;
> > >
> > > `Real to Complex Conversion`
> > >
> > > *Additional Discussion:*
> > >
> > > Thank you for finding the discussion valuable. We will make sure to include it in the camera-ready version.
> > >
> > > &nbsp;
> > >
> > > *Independence:*
> > >
> > > Thank you for pointing this out and helping us refine our explanation. We acknowledge that we overlooked considering the polar representation of our transform. However, it is important to highlight an intriguing aspect: while magnitudes are related, the phases are entirely independent, which has significant implications. Prior works [1] and [2] have demonstrated how phase information can be effectively leveraged for object discovery and image classification tasks, respectively.
> > >
> > > This phase independence could potentially explain why we observe improved performance with our iRGB color model. Accordingly, we would like to revise our earlier statement on independence to the following: our transform produces components that are phase-wise independent.
> > >
> > > &nbsp;
> > >
> > > `Binary Segmentation`
> > >
> > > *Mapping:*
> > >
> > > As mentioned earlier, our goal was to design an end-to-end complex-valued neural network, so it obviously performs complex-to-complex mapping. To enable other types of mappings, we introduced the R2C transform at the input and the (1, i) encoding at the output, as pre-processing and post-processing steps, respectively.
> > >
> > > Our primary motivation for developing this end-to-end complex-valued neural network was to unlock the full potential of complex-valued architectures in the computer vision domain.
> > >
> > > &nbsp;
> > >
> > > *Background as $i$:*
> > >
> > > This was intended as an analogy. Just as the foreground and background in an image are physically separate and distinct, we can treat them as mutually independent by representing one as the real part and the other as the imaginary part. Therefore, if $y$ denotes the usual binary segmentation map (i.e. foreground=1 & background=0) highlighting the foreground, then $1-y$ becomes the map highlighting the background. Representing these as the real and imaginary parts results in a complex-valued target map, $y′=y+i(1−y)$.
> > >
> > > In this formulation, since $y=1$ for foreground pixels, all foreground pixels in $y′$ are represented by $1$. Similarly, since $y=0$ for background pixels, all background pixels in $y′$ are represented by $i$. As far as inference in concerned, if the output is a complex map $a+ib$, we compute the final output map in the real domain by taking the average of $a$ and $1−b$ maps. We hope this explanation clarifies why our $(1, i)$ encoding of the segmentation map is intuitive.
> > >
> > >
> > >
> > > &nbsp;
> > >
> > > *References:*
> > >
> > > [1] Löwe, S., Lippe, P., Rudolph, M., & Welling, M. (2022). Complex-valued autoencoders for object discovery. Transactions on Machine Learning Research.
> > >
> > > [2] Chen, G., Peng, P., Ma, L., Li, J., Du, L., & Tian, Y. (2021). Amplitude-phase recombination: Rethinking robustness of convolutional neural networks in frequency domain. In Proceedings of the IEEE/CVF International Conference on Computer Vision (pp. 458-467).

---

> > > > ### Author Response · Authors · 2024-12-03
> > > > **Discussion Follow-up (for Reviewer F89z)**
> > > >
> > > > Dear Reviewer F89z,
> > > >
> > > > Thank you very much for your thoughtful and constructive review of our paper. We truly appreciate the time and effort you have invested in providing us with valuable feedback. Your insights have been instrumental in helping us explain our work better. It was an absolute pleasure to have you as a reviewer.
> > > >
> > > > Based on our discussion, we believe we have addressed the concerns you raised and clarified any points of confusion. We hope these clarifications have resolved the key issues and align with your expectations.
> > > >
> > > > Given that the other reviewers have rated the paper positively, we kindly request you to reconsider your current rating in light of these discussions. We believe that a consensus among reviewers could be reached and would be grateful for your updated evaluation.
> > > >
> > > > Thank you once again for your time and for contributing to this review process.
> > > >
> > > > &nbsp;
> > > >
> > > > Warm regards,
> > > >
> > > > Authors of #1276

---

### Official Review · Reviewer_BUvX · 2024-11-02

**Soundness:** 3
**Presentation:** 2
**Contribution:** 3
**Rating:** 6
**Confidence:** 5

**Summary:**

This paper presents a transformation to map RGB images into complex domain and an associate network comprising a loss function to handle such complex inputs.

Overall, the contributions are significant and may be of interest to the community, but the paper organization could be improved and more focus should be placed on the transformation.

**Strengths:**

1) The proposed transformation is novel and may be an important contribution for the community working in complex and hypercomplex domains.
2) Although not novel, using the fourier filter module is good to handle complex inputs.

**Weaknesses:**

1) While the method sounds, the results are not impressive and surely they are not statistically significant. As of my experience, complex, quaternion, and in general hypercomplex models clearly outperform real-valued counterparts when they are able to catch some underlying physical process intrinsic into data. Maybe, the tasks chosen by the authors do not highlight the effectiveness of their method. Maybe, the authors could stress more the parameters saving of using a complex model with respect to a real-valued one, which can help reducing the computational load while obtaining comparable results.
2) The authors should have focused more on the transformation, which is a novel contribution, and better show its properties (see questions).
3) Some key references to related works are missing, the authors should at least give credit to them, or better try to compare their method with them. Some of them follow, but I encourage the authors to better explore previous literature on complex, quaternion and hypercomplex networks.

[1] C. Trabelsi, O. Bilaniuk, Dmitriy Serdyuk, Sandeep Subramanian, J. F. Santos, Soroush Mehri, Negar Rostamzadeh, Yoshua Bengio, C. Pal, "Deep Complex Networks", ICLR 2017.

[2] E. Grassucci, A. Zhang, D. Comminiello, "PHNNs: Lightweight Neural Networks via Parameterized Hypercomplex Convolutions", IEEE Transactions on Neural Networks and Learning Systems, (Volume: 35, Issue: 6, June 2024).


Minor comments:

The Saxon Genitive should be avoided in scientific writing, although I know that both ChatGPT and Grammarly insert it. I suggest the authors to remove all the Saxon genitives in the paper.

**Questions:**

1) I am very curious about the invertibility of the proposed transform. Given the Algorithm 2 in Appendix A, would it be possible to have some experiments to prove its effectiveness? I think that this transformation is the real contribution of the paper, as it allows a direct mapping between greyscale and RGB images, which was lacking in complex and quaternion papers that often struggle to do so.
2) Which is the computational load in terms of FLOPs, runtime memory, and time of the proposed model?

---

> ### Author Response · Authors · 2024-11-22
> **Author response to reviewer BUvX (part 1)**
>
> ## Weakness-1 [part 1 of 2]: Experiments (Computational efficiency + Comparison with real-valued counterpart) and Backbone necessity
>
> - The main motivation of this paper was to build the first-of-its-kind binary segmentation model entirely in the complex domain, and therefore we took up the tasks such as SOD, shadow detection, blur detection and foreground extraction (for complex-valued images). While we have clearly outperformed the existing works on complex-valued neural networks in each of these tasks (see Tables 2-5; as acknowledged by reviewer B2bV as well), our results are currently comparable to state-of-the-art methods that operate exclusively in the real domain.
>
>
> - As you rightly pointed out, complex-valued networks are usually efficient on computational front, we tried to collect the number of parameters of  as many methods as possible and have reported them now in the manuscript. As expected, we can clearly notice that our number of parameters is on the lower side while obtaining comparable results. We appreciate your valuable feedback, which helped highlight this important aspect of our work.
>
> - We would like to emphasize that the competing real-valued methods cannot be considered direct counterparts to our approach, as their network architectures differ significantly. To ensure a fair comparison, we adapted our models by converting their parameters to real-valued ones. Our results are indeed better than the real-valued counterparts, as can be seen in the tables below:
>
>
> For Salient Object Detection:
> |   Dataset |     |    DUTS    |           |         |  |  HKU-IS         |           |         | |  ECSSD         |           |         |
> |---------------|---------|--------|-----------|---------|---------|--------|-----------|---------|---------|--------|-----------|---------|
> |    | $S_m \uparrow$ |  $maxF\uparrow$ |  $E_{\xi}^{max}\uparrow$ |  $MAE\downarrow$ | $S_m \uparrow$ |  $maxF\uparrow$ |  $E_{\xi}^{max}\uparrow$ |  $MAE\downarrow$ |$S_m \uparrow$ |  $maxF\uparrow$ |  $E_{\xi}^{max}\uparrow$ |  $MAE\downarrow$ |
> | Real-DCSNet | 0.864 | 0.832 | 0.912 | 0.043 | 0.908 | 0.920 | 0.948 | 0.041 | 0.907 | 0.912 | 0.956 | 0.034 |
> | DCSNet | **0.894** | **0.874** | **0.941** | **0.039** | **0.927** | **0.945** | **0.960** | **0.034** | **0.917** | **0.924** | **0.967** | **0.029** |
>
>
>
>
> | Dataset |  | PASCAL-S    |           |         |    | DUT-O       |           |         |
> |---------|---------|:-----------:|-----------|---------|---------|--------|-----------|---------|
> |  |$S_m \uparrow$ |  $maxF\uparrow$ |  $E_{\xi}^{max}\uparrow$ |  $MAE\downarrow$ |$S_m \uparrow$ |  $maxF\uparrow$ |  $E_{\xi}^{max}\uparrow$ |  $MAE\downarrow$ |
> |Real-DCSNet| 0.847 | 0.841 | 0.896 | 0.069 | 0.817 | 0.747 | 0.851 | 0.062 |
> |DCSNet| **0.866** | **0.850** | **0.903** | **0.062** | **0.839** | **0.776** | **0.880** | **0.056** |
>
> &nbsp;
>
> For Defocus Blur Detection:
>
> | Method | DUT | | CUHK | |
> |------|-----|-----|-----|-----|
> |        | $\mathcal{F}_{\beta} \uparrow$ | MAE$\downarrow$ | $\mathcal{F}_{\beta} \uparrow$ | MAE$\downarrow$ |
> | Real-DCSNet | 0.841| 0.121 | 0. 899 | 0.064 |
> | DCSNet | **0.894**| **0.058** | **0.907** | **0.045**|
>
> &nbsp;
>
> For Shadow Detection:
> | Method | ISTD | SBU |
> |-----|-------|-------|
> |  | BER $\downarrow$ | BER $\downarrow$ |
> | Real-DCSNet | 1.83 | 3.45 |
> | DCSNet | **1.49** | **3.05** |
>
> &nbsp;
>
> For InSAR Foreground Extraction:
>
> | Method | mIoU $\uparrow$|
> |-------|---------|
> |Real-DCSNet | 0.85 |
> |DCSNet| **0.89** |
>
> *Note: **Bold** indicates better result.*
>
>
> - In order to develop complex-valued binary segmentation networks, we required a backbone (encoder) trained on a large dataset such as ImageNet, as no pre-trained token-based complex-valued network currently exists. Therefore, we trained such a network from scratch and reported the results in Table 1(a). Interestingly, our network outperformed existing complex-valued networks, motivating us to evaluate it further on complex-valued datasets. As shown in Table 1(b), our method demonstrated superior performance in those evaluations as well.
>
> - In Table 1(a), we compared with some relevant and well-known real-valued networks too as baselines. While we acknowledge the existence of more advanced image classification networks, our goal was not to achieve state-of-the-art performance in image classification. We just needed a decent token-based complex-valued network to serve as backbone to our main network, and we believe we were able to achieve this, as the results were comparable with the baselines chosen.

---

> ### Author Response · Authors · 2024-11-22
> **Author response to reviewer BuvX (part 2)**
>
> ## Weakness-1 [part 2 of 2]: Average SOD performance
> Below, we compare the average performance across the five SOD datasets presented in Table 2. While our method achieves the best performance in terms of the maxF metric, it consistently ranks at least second-best across all metrics—a distinction no other method in the comparison achieves.
>
> | Method | $S_m \uparrow$ | $maxF\uparrow$ | $E_{\xi}^{max} \uparrow$ | $MAE\downarrow$ |
> |--------|--------|--------|--------|--------|
> |PiCANet     | 0.871 | 0.854 | 0.913 | 0.046 |
> |BASNet      | 0.872 | 0.856 | 0.909 | 0.051 |
> |PoolNet     | 0.879 | 0.864 | 0.914 | 0.049 |
> |EGNet-R     | 0.886 | 0.868 | 0.918 | 0.048 |
> |MINet-R     | 0.885 | 0.868 | 0.919 | 0.046 |
> |LDF-R       | 0.892 | 0.876 | 0.921 | `0.043` |
> |CSF-R2      | 0.892 | 0.874 | 0.911 | 0.049 |
> |GateNet-R   | 0.888 | 0.874 | 0.922 | 0.047 |
> |VST         | `0.904` | **0.894** | `0.932` | 0.045 |
> |FCCN        | 0.824 | 0.813 | 0.878 | 0.078 |
> |SCVUNet     | 0.831 | 0.827 | 0.893 | 0.069 |
> |DCSNet (Ours) | **0.893** | `0.895` | **0.930** | **0.044**|
>
> *Note: `Red` indicates best result, and **bold** indicates second best result.*

---

> ### Author Response · Authors · 2024-11-22
> **Author response to reviewer BUvX (part 3)**
>
> ## Weakness-2 & Question-1: Further analysis of R2C transform and its invertibility
>
>
> - We appreciate the acknowledgment of the novelty and advantages of our RGB-to-iRGB (R2C) transform, including its invertibility and the ability to visualize its components as grayscale images. To demonstrate its effectiveness, we originally included a demo in the supplementary material at the time of submission. This demo showcased two sample images, which were converted into our complex color domain (iRGB) using the R2C transform and then reconstructed back to the original images using the inverse-R2C transform.
>
> - Following your suggestion for further analysis, we have now provided additional examples and included SSIM scores in pdf (R2C_demo.pdf) to quantitatively measure the similarity between the original and reconstructed images. These results, presented in the updated supplementary material, consistently show an SSIM score of 1, thereby validating our claim that the R2C transform is perfectly invertible.
>
>
> - Our R2C transform is not only novel but also provides some additional benefits compared to the existing complex-valued color models and encodings. For example, quaternion neural networks [1] have RGB image inputs in the form of $0 + iR + jG + kB$ to create a quaternion representation, which artificially forces the real part to be zero. Similarly, CDS [3] proposes to uses Lab color space to generate complex input as: $L+i0$ and $a+ib$. Here, the imaginary component of the first complex channel is also artificially set to zero. In contrast, in our case, we managed to derive complex channels naturally by locating colors on the two kinds of argand planes we discovered in the RGB color space.
>
> - In a slightly different approach, DCN [2] generated complex input $I + i f(I)$ from an image $I$ by using a convolutional block f. The idea is to learn imaginary part from the real part itself. However, this introduces dependency between the real and imaginary components, which should not be the case. In our case, however, the real and imaginary components are derived from the orthogonal components of a vector, which ensures they are independent of each other.
>
>
> [1] Zhu, X., Xu, Y., Xu, H., & Chen, C. (2018). Quaternion convolutional neural networks. In Proceedings of the European conference on computer vision (ECCV)
>
> [2] C. Trabelsi, O. Bilaniuk, Dmitriy Serdyuk, Sandeep Subramanian, J. F. Santos, Soroush Mehri, Negar Rostamzadeh, Yoshua Bengio, C. Pal, "Deep Complex Networks", ICLR 2017.
>
> [3] Singhal, U., Xing, Y., & Yu, S. X. (2022). Co-domain symmetry for complex-valued deep learning. In Proceedings of the IEEE/CVF Conference on Computer Vision and Pattern Recognition.

---

> > ### Author Response · Authors · 2024-11-22
> > **Author response to reviewer BUvX (part 4)**
> >
> > ## Weakness 3: Missed references and comparisons
> >
> > Thank you for suggesting the missing references. We agree that these works should have been discussed. Our focus was on binary segmentation, and we inadvertently overlooked them as they primarily address image classification. To address this, we have included a comparison of our encoder with these and other complex-valued works on the ImageNet and CIFAR-10 datasets in the table below. The results clearly demonstrate that our method performs better.
> >
> >
> > | Model | PHNN (ResNet50) | DCN | FCCN | DCSNet encoder|
> > |-----|------|-------|--------|--------|
> > | Acc (\%) | 68.6 |  72.64 | 77.27 | **78.83**|
> >
> >
> > | Model | PHNN (ResNet152) | DCN | FCCN | DCSNet encoder |
> > |-------|------|------|------|----------------|
> > | Acc (\%)| 90.5 | 89.6 | 93.6 |   **94.3**      |
> >
> > *Note: **Bold** indicates better result.*
> >
> >
> > Below, we briefly summarize these works:
> >
> > - DCN [1] introduced the foundational elements of complex-valued neural networks, demonstrating their utility in image classification tasks. FCCN [3] extended this by designing fully complex-valued networks through the use of complex convolutions across all layers. More recently, PHNN [2] proposed a generalized framework for hypercomplex networks, introducing parameterized hypercomplex convolutional layers that learn convolution rules from data using the Kronecker product. This approach provides flexibility for handling complex, hypercomplex, and quaternion representations.
> >
> > - While these prior works predominantly address image classification in the complex domain, our work uniquely focuses on binary segmentation in the complex domain. Furthermore, unlike FCCN's convolution-based design for developing end-to-end complex-valued networks, our approach leverages a token-based methodology to achieve the same goal.
> >
> >
> > [1] C. Trabelsi, O. Bilaniuk, Dmitriy Serdyuk, Sandeep Subramanian, J. F. Santos, Soroush Mehri, Negar Rostamzadeh, Yoshua Bengio, C. Pal, "Deep Complex Networks", ICLR 2017.
> >
> > [2] E. Grassucci, A. Zhang, D. Comminiello, "PHNNs: Lightweight Neural Networks via Parameterized Hypercomplex Convolutions", IEEE Transactions on Neural Networks and Learning Systems, (Volume: 35, Issue: 6, June 2024).
> >
> > [3] Saurabh Yadav; Koteswar Rao Jerripothula, FCCNs: Fully Complex-valued Convolutional Networks using Complex-valued Color Model and Loss Function. ICCV 2023.

---

> > > ### Author Response · Authors · 2024-11-22
> > > **Author response to reviewer BUvX (part 5)**
> > >
> > > ## Question 2 & Weakness 4: Computational load & Saxon Genitive
> > >
> > > ### 1. Computational load
> > >
> > > Thank you for the suggestion. In the table below, we present a comparison of FLOPs, GPU memory usage, and inference time (for a batch size of 1) between our proposed DCSNet and FCCN (ResNet152), the previous state-of-the-art complex-valued network on ImageNet.
> > >
> > > For Classification:
> > > |  | FCCN | DCSNet|
> > > |---|------|-------|
> > > |gFLOPS $\downarrow$ | 28 | **6** |
> > > |GPU memory (MB) $\downarrow$| 330 | **132**|
> > > | Time (ms) $\downarrow$| 61 | **13** |
> > >
> > > &nbsp;
> > >
> > >
> > > For Binary Segmentation:
> > > | | FCCN | DCSNet|
> > > |---|------|-------|
> > > | gFLOPS $\downarrow$ | 59 | **22** |
> > > |GPU memory (MB) $\downarrow$| 573 | **289** |
> > > | Time (ms) $\downarrow$| 84 | **30** |
> > >
> > > *Note: **Bold** indicates better result.*
> > >
> > > &nbsp;
> > > ### 2. Saxon Genitive
> > >
> > > We are thankful for your valuable suggestion and feedback. We have removed all Saxon genetives from the revised manuscript.

---

> > > > ### Comment · Reviewer_BUvX · 2024-12-01
> > > > **Thanks for the reply**
> > > >
> > > > I would like to thank the authors for their effort and the quality of the response.
> > > > I find the results on invertibility, the additional comparisons, and the computational load metrics very interesting.
> > > > However, I could not find any of these interesting discussions or results in the main paper and this is a little bit disappointing as these results could have really strengthened authors claims and results.
> > > > I would raise my score by one point since the results are pretty interesting, but not more than this since the paper lacks those discussions and results.

---

> ### Author Response · Authors · 2024-12-02
> **Incorporating Missing Results in Camera-ready Version**
>
> Thank you for your thoughtful review and for raising the score. We deeply appreciate your recognition of the additional results and discussions we provided in our response, including those on invertibility, comparisons, and computational load metrics. We apologize for not including these in the main paper; this omission was unintentional and due to unforeseen circumstances related to the lead author's health.
>
> Our priority during the rebuttal phase was to address reviewers' concerns as thoughtfully and comprehensively as possible, which left us unable to integrate these results into the manuscript in time. However, we fully agree that these findings significantly strengthen our claims and results. We will ensure they are thoroughly incorporated into the camera-ready version, along with additional context and discussion, to maximize the paper's clarity and impact.
>
> Thank you again for your constructive feedback and for recognizing the value of these results.

---

### Comment · Area_Chair_nR2G · 2024-11-25

Dear reviewers,

A reminder that **November, 26** is the last day to interact with the authors, before the private discussion with the area chairs. At the very least, please acknowledge having read the rebuttal (if present). If the rebuttal was satisfying, please improve your score accordingly. Finally, if you have concerns that might be solved shortly, please exploit the remaining time.

Thanks,
The AC

---

### Author Response · Authors · 2024-12-04
**Summary and Common Response**

We sincerely thank the reviewers for their valuable feedback and constructive discussions. We are encouraged by the acknowledgment of the novelty and contributions of our work.

&nbsp;

## Paper Summary:

This paper addresses a long-standing limitation in applying complex-valued neural networks to computer vision tasks while solving binary image segmentation entirely within the complex domain. Our work builds on the foundational principles of the FCCN [2] paper but extends the field significantly by introducing:

- **iRGB Color Model:** A novel, invertible complex-valued color model, enabling complex-valued processing of RGB images. Our experiments show iRGB surpasses the iHSV color model (proposed by FCCN [2]), providing a robust foundation for future research.

- **DCSNet:** A novel token-based complex-valued architecture exploring the frequency domain using complex Fourier transform and a multi-scale encoder-decoder structure. This approach learns rich complex-valued representations and achieves SOTA performance (while using complex-valued neural networks) across multiple binary image segmentation tasks.

- **Complex Encoding for Segmentation Targets:** A unique (1, i) encoding method for binary segmentation maps, inspired by the physical independence of foreground and background elements, aligning seamlessly with the mathematical properties of complex numbers.

Beyond achieving state-of-the-art results on real-valued and complex-valued datasets, our work broadens the application of complex-valued neural networks to more computer vision tasks. For instance, our invertible R2C transform further opens opportunities in image synthesis and generation tasks, demonstrating the transformative potential of complex-valued representations.


&nbsp;


## Review & Discussion Summary:

**Reviewer BUvX:**
BUvX acknowledged the potential of complex-valued networks but raised concerns about comparisons and additional analyses. We provided detailed comparisons with real-valued counterparts, computational efficiency results for DCSNet, and further insights into our R2C transform. While our responses appeared satisfactory, the reviewer was disappointed these updates were not included in the revised manuscript. We acknowledge this oversight and will include them in the camera-ready version.

**Reviewer F89z:**
F89z recognized the novelty of our R2C transform but sought clarification on its advantages over existing real-to-complex transforms, including simpler alternatives. We provided detailed explanations and quantitative comparisons to highlight its benefits, addressing these points thoroughly. F89z was further interested in the advantages offered over the trivial [R+Gi,G+Bi] transform, and we explained them too. Upon F89z’s suggestion, we revisited one of the advantages and revised it too.

Additionally, F89z requested insights into the improvements our DCSNet offers over DCN [1]. We clarified that our approach remains fully within the complex domain, unlike DCN [1], which does not solve computer vision tasks entirely in the complex domain. F89z also raised questions about local feature learning and our (1,i) encoding, which we addressed in detail.

While we have responded to all concerns of F89z, we have not yet received further communication regarding re-evaluation of the score.

**Reviewer QqAh:**
QqAh sought clarity on the differences between our work and FCCN [2] and requested further details on R2C and architectural choices. We addressed all queries, providing detailed explanations and additional experimental results. The reviewer was satisfied but suggested exploring wavelet filters in future work.

**Reviewer B2bV:**
B2bV was highly positive, awarding a score of 8, and requested clarification on result interpretation. We highlighted how our method surpassed 90% performance and achieved a 2.86% improvement over prior complex-valued approaches.


&nbsp;

`FINAL REMARK:` We believe this paper is a significant step toward realizing the full potential of complex-valued neural networks in computer vision. We hope the advancements made, their implications and the review discussions will be carefully considered while making the final decision.

&nbsp;

Warm Regards,

Authors of #1276

---

### Meta-Review · Area_Chair_nR2G · 2024-12-13

**Metareview:**

The paper considers binary segmentation with a complex-valued neural network. It introduces several novel components, including (a) a real-to-complex transformation of the RGB values, (b) a Fourier module, and (c) a segmentation loss in the complex domain. Of the 3 reviews with confidence > 2, the paper had a negative review round with all 3 reviewers recommending rejection. After the rebuttal phase, 1 reviewer is still highly critical of the work, while the other 2 have retained a marginal accept score.

In general, there are several issues that were not fully clarified in the rebuttal phase. In particular, (a) the novelty of the method is not clear, especially since the authors have decided to focus on a narrow computer vision use case; (c) the results when comparing to a real-valued scenario are not convincing; (c) many experiments were not included in the main paper, and (d) some reviewers claim that a more complete mathematical analysis of the conversion is missing.

On my side, I believe the paper is interesting for the community working on complex-valued models, but the narrow scope and unconvincing rebuttal makes me lean towards a rejection. I believe the paper can be significantly strengthened by the inclusion of the comments that were discussed in the rebuttal phase, but this would require a significant amount of work that could also change the structure or claims of the paper.

**Additional Comments On Reviewer Discussion:**

- **Reviewer B2bV**: they claimed to have limited experience in the field, and they haven't interacted during the rebuttal phase. Their score significantly differed from the other reviewers. *I ignored most of the review for the final decision.*

- **Reviewer QqAh**: they were concerned mostly about the limited novelty of the approach, and about a lack of evaluation of competing conversion approaches (such as complex wavelets). During the rebuttal they improved the score from a 5 to a 6, but still leaning towards a rejection due to the limited novelty.

- **Reviewer F89z**: they highlighted multiple concerns, including the lack of novelty, the limited mathematical analysis of the solution, and the unconvincing results. The authors provided a lengthy rebuttal, but the reviewer remained unconvinced on most points and keeps recommending rejection. *This review and the reviewer's comments have influenced heavily my final decision*, and other reviewers agreed with F89z in the final discussion.

- **Reviewer F89z** had similar concerns with respect to F89z (unconvincing results, limited novelty, limited analysis of the methods). While the rebuttal was more convincing (and they increased the score from 5 to 6), they remain concerned that most of the analysis shown in the rebuttal does not appear in the main paper. *I agree with this evaluation*, as reflected in the metareview.

---

### Decision · Program_Chairs · 2025-01-22

Reject